# Off-Policy Corrected Reward Modeling
# for Reinforcement Learning from Human Feedback

**Johannes Ackermann**
The University of Tokyo and RIKEN AIP
ackermann@ms.k.u-tokyo.ac.jp

**Takashi Ishida**
RIKEN AIP and The University of Tokyo

**Masashi Sugiyama**
RIKEN AIP and The University of Tokyo

## Abstract

Reinforcement Learning from Human Feedback (RLHF) allows us to train models, such as language models (LMs), to follow complex human preferences. In RLHF for LMs, we first train an LM using supervised fine-tuning, sample pairs of responses, obtain human feedback, and use the resulting data to train a reward model (RM). RL methods are then used to train the LM to maximize the reward given by the RM. As training progresses, the responses generated by the LM no longer resemble the responses seen by the RM during training, leading to the RM becoming inaccurate. The score given by the RM keeps increasing, but the learned behavior no longer matches the human preferences. This issue is known as overoptimization. We investigate overoptimization from the point of view of distribution shift and show that the shift results in an inconsistent estimate of the RM parameters, leading to an inconsistent estimate of the policy gradient. We propose Off-Policy Corrected Reward Modeling (OCRM), which iteratively off-policy corrects the RM using importance weighting, without requiring new labels or samples. This results in a more accurate RM, which empirically leads to an improved final policy. We validate our approach in experiments with summarization and chatbot datasets and show that it performs significantly better than standard RLHF methods and baselines.

## 1 Introduction

Large language models (LLMs) have had a large impact on society through the availability of chatbots such as ChatGPT (Ouyang et al., 2022). It is important to ensure that these LLMs behave in a desired way and thus Reinforcement Learning from Human Feedback (RLHF) (Christiano et al., 2017) has become an important tool to align pre-trained language models (LMs) with human preferences. Given a pretrained LM, RLHF typically consists of three steps (Stiennon et al., 2020): Firstly, supervised fine-tuning (SFT), in which the LM is trained to imitate example input-output pairs. Secondly, pairs of replies are sampled from the SFT model, labeled by humans according to their preferences, and collected in a dataset which is then used to train a reward model (RM). Finally, starting from the SFT model, reinforcement learning (RL) is used to train the RL model to optimize the score given by the RM. While the training progresses, the RL model becomes increasingly different from the SFT model. As the RM has only been trained on replies provided by the SFT model, the RM can no longer provide accurate rewards. The output of the RM keeps increasing, but the quality of the responses as judged by a human stagnates or even decreases. This phenomenon is known as Goodharting or overoptimization (Gao et al., 2023). We investigate overoptimization as a distribution shift problem, caused by the difference between the RL model and the SFT model, whose outputs the RM is trained on. While previous work (Gao et al., 2023; Song et al., 2024; Fluri et al., 2025) has discussed that overoptimization can be considered a consequence of distribution shift, to our knowledge, the applicability of methods from the distribution shift adaptation literature has not been investigated.

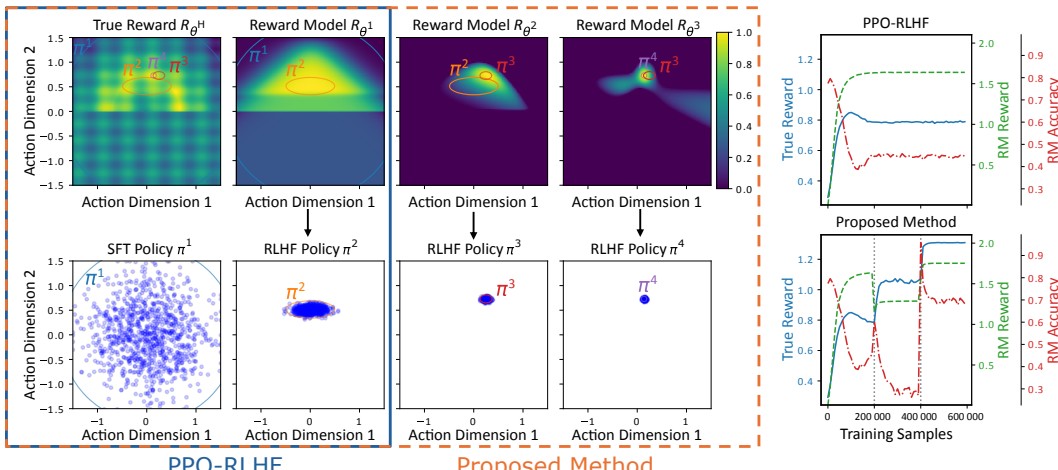

Figure 1: Visualization of our approach in a 2D stateless task. On the top-left side, the true reward function and learned RMs are visualized, on the lower-left side the SFT policy and learned policy distributions are shown. The colored circles contain 99% of each policy's action samples. The first RM $R_{\theta 1}$ is trained on data sampled from the SFT policy $\pi^1$ and thus performs well on pairs of actions sampled from $\pi^1$, but is inaccurate on samples generated by $\pi^2$. This leads the policy trained with standard PPO-RLHF to stagnate early (top-right). Using IW, after $k = 200,000$ samples, we train an off-policy corrected RM $R_{\theta 2}$ that is accurate on samples generated by $\pi^2$ and can thus be used to continue training and obtain a better policy $\pi^3$. Iterating this process, we obtain a better final policy (bottom-right).

We investigate this phenomenon and show that standard RLHF methods (Stiennon et al., 2020; Ahmadian et al., 2024) provide an inconsistent estimate of the RM parameters for all but the first policy update, and therefore generally also provide an inconsistent policy gradient estimate. They thus generally do not converge to the optimal policy, even with unlimited available data and unlimited training iterations. We propose a method that corrects for this distribution shift by using importance weighting (IW) (Shimodaira, 2000), called Off-Policy Corrected Reward Modeling (OCRM). While using the resulting corrected RM to estimate each policy gradient would be desirable, obtaining it requires the RM to be retrained after each policy update and is thus not computationally feasible. We therefore propose to approximate this method with multiple training stages, each consisting of training a corrected RM with IW and then performing multiple policy updates. As illustrated in Figure 1, our approach iteratively improves the accuracy of the RM on samples of the current policy and converges to a higher final reward, without any additional data, while standard Proximal Policy Optimization (PPO)-RLHF (Stiennon et al., 2020) stagnates.

Our main contributions are as follows: We provide an analysis of overoptimization in RLHF as distribution shift caused by the change of the policy due to RL updates, showing that even with unlimited training data standard RLHF may not converge to an optimal policy. Secondly, we propose to address this issue by using IW, giving us a consistent estimate of the RM parameters, and provide an estimation error bound for our proposed method. Finally, we implement our method and show that it performs well on two LM alignment tasks: In both the TL;DR summarization task (Stiennon et al., 2020) and the Alpaca-Farm chatbot task (Dubois et al., 2023), our method improves the alignment performance, outperforming standard PPO-RLHF (Stiennon et al., 2020), Direct Preference Optimization (DPO) (Rafailov et al., 2023) and related baselines (Zhou et al., 2024; Lang et al., 2024).

## 2 Related work

The most closely related method is Weighted Preference Optimization (WPO) (Zhou et al., 2024). They improved DPO by multiplying the DPO loss with a weight term, similar to an

importance weight. However, they used the probability under the current policy as weight and thus can only provide a correct off-policy correction if the dataset was generated by a uniform SFT model, as we show in Appendix B. While they focus on DPO, we focus on policy gradient methods. Also closely related is Reward Learning on Policy (RLP) (Lang et al., 2024). RLP retrains the RM with samples from the current RL model, using either an unsupervised representation learning method or a pseudo-labeling approach. Unlike RLP, we show that our method provides a consistent estimate of the new RM parameters. We empirically compare our approach with both WPO and RLP and provide an extended discussion of the related work in Appendix C.

## 3 Background

In this section, we will explain the background for RLHF and IW. We denote the state $s$, action $\mathbf{a}$ and policy $\pi : \mathcal{S} \to \mathrm{Dist}(\mathcal{A})$, with state set $\mathcal{S}$, action set $\mathcal{A}$, and distribution $\mathrm{Dist}(\cdot)$. For clarity, we focus on the case of LMs and thus states $s$ correspond to prompts and actions $\mathbf{a}$ correspond to completions by an LM. $\mathcal{S}$ and $\mathcal{A}$ are sets of sequences of tokens.

### 3.1 Reinforcement learning from human feedback

Following the setup of Stiennon et al. (2020), RLHF consists of three main steps: SFT, reward modeling, and RL.

**Supervised fine-tuning** As policy we consider an autoregressive model $\pi(\mathbf{a} \mid s) = \prod_t \pi(a_t \mid s, a_{<t})$, where $a_t$ is the $t$-th token, $a_{<t}$ refers to the response tokens before $a_t$ and $\mathbf{a}$ refers to the entire response. First, a pretrained (PT) LLM $\pi^{\mathrm{PT}}$ is fine-tuned on a given dataset $D_{\mathrm{SFT}} = \{(s^j, \mathbf{a}^j)\}_{j=1}^{N_{\mathrm{SFT}}}$. We denote the resulting policy as SFT model $\pi^1$.

**Reward modeling** Given a prompt $s \sim P(s)$, where $P(s)$ is the distribution over prompts, two completions $\mathbf{a}_0, \mathbf{a}_1$ are sampled from the SFT model $\pi^1(\mathbf{a} \mid s)$. A human annotator then selects their preferred reply, according to their personal preferences, giving us a dataset $D_{\mathrm{RM}} = \{(s^j, \mathbf{a}_{\mathrm{w}}^j, \mathbf{a}_{\mathrm{l}}^j)\}_{j=1}^{N_{\mathrm{RM}}} \overset{\mathrm{i.i.d.}}{\sim} P_{\pi^1}(s, \mathbf{a}_{\mathrm{w}}, \mathbf{a}_{\mathrm{l}})$, where $\mathbf{a}_{\mathrm{w}}$ and $\mathbf{a}_{\mathrm{l}}$ are the preferred (winning) and not preferred (losing) replies, respectively. The Bradley-Terry (BT) model (Bradley & Terry, 1952) is used as a modeling assumption that states that the probability of preferring one option $\mathbf{a}_1$ over another option $\mathbf{a}_0$ is the logistic function $\sigma(x) = 1/(1 + e^{-x})$ of the difference of an (unknown) scalar reward $R : \mathcal{S} \times \mathcal{A} \to \mathbb{R}$, i.e.,

$$P(\mathbf{a}_1 > \mathbf{a}_0 \mid s) = \sigma\left(R(s, \mathbf{a}_1) - R(s, \mathbf{a}_0)\right). \tag{1}$$

This assumption is used to train an RM $R_\theta : \mathcal{S} \times \mathcal{A} \to \mathbb{R}$ with parameters $\theta \in \Theta$ by minimizing the cross-entropy loss based on the BT model:

$$\mathcal{L}_{\mathrm{RM}}^{\pi^1}(\theta) = -\mathbb{E}_{(s, \mathbf{a}_{\mathrm{w}}, \mathbf{a}_{\mathrm{l}}) \sim P_{\pi^1}}[\log \sigma(R_\theta(s, \mathbf{a}_{\mathrm{w}}) - R_\theta(s, \mathbf{a}_{\mathrm{l}}))]. \tag{2}$$

We denote the per-sample loss as $l_{\mathrm{RM}}(s, \mathbf{a}_{\mathrm{w}}, \mathbf{a}_{\mathrm{l}}; \theta) = -\log \sigma(R_\theta(s, \mathbf{a}_{\mathrm{w}}) - R_\theta(s, \mathbf{a}_{\mathrm{l}}))$ and in practice use an empirical estimate of the reward parameters obtained by minimizing the sample average $\hat{\theta} = \arg\min_{\theta \in \Theta} 1/N_{\mathrm{RM}} \sum_{j=1}^{N_{\mathrm{RM}}} l_{\mathrm{RM}}(s^j, \mathbf{a}_{\mathrm{w}}^j, \mathbf{a}_{\mathrm{l}}^j; \theta)$. We denote the parameters of the true "human" RM as $\theta^{\mathrm{H}}$, which is generally not in our hypothesis class $\theta^{\mathrm{H}} \notin \Theta$.

**Reinforcement learning** As is common in the RLHF literature (Shao et al., 2024; Ahmadian et al., 2024), we consider an episode consisting of a single state $s$ and action $\mathbf{a}$. RL is used to train the policy $\pi_\phi^2$, with parameters $\phi$ initialized from the SFT model $\pi^1$, to maximize the expected return $J(\phi, \theta) = \mathbb{E}_{s \sim P(s), \mathbf{a} \sim \pi_\phi}[R_\theta'(s, \mathbf{a})]$, with the regularized reward

$$R_\theta'(s, \mathbf{a}) = R_\theta(s, \mathbf{a}) - \beta D_{\mathrm{KL}}(\pi_\phi^2(\cdot \mid s) || \pi^1(\cdot \mid s)).$$

$D_{\mathrm{KL}}$ is the Kullback-Leibler (KL) divergence with parameter $\beta > 0$ controlling the strength of the regularization. This KL-penalty keeps the model close to the training distribution of

the RM, preventing extreme overoptimization (Gao et al., 2023). The expected return is then optimized using an advantage-based policy gradient (PG) method (Williams, 1992),

$$\nabla_\phi J(\phi, \theta) = \mathbb{E}_{(s,\mathbf{a}) \sim P_\pi} \left[ A_\theta^\pi(s, \mathbf{a}) \nabla_\phi \log \pi_\phi(\mathbf{a}|s) \right], \tag{3}$$

with the advantage function $A_\theta^\pi(s, \mathbf{a}) = R_\theta'(s, \mathbf{a}) - V_\theta^\pi(s)$ and value function $V_\theta^\pi(s) = \mathbb{E}_{\mathbf{a} \sim \pi}[R_\theta'(s, \mathbf{a})]$. PG-based RLHF methods differ in the updates and advantage estimation methods they use. PPO-RLHF (Stiennon et al., 2020) uses Proximal Policy Optimization (PPO) (Schulman et al., 2017), which implements a proximal update and uses Generalized Advantage Estimation (Schulman et al., 2016). Group Relative Policy Optimization (GRPO) (Shao et al., 2024) instead uses Monte Carlo estimates of the advantage in each state by evaluating multiple actions. We sometimes omit $\phi$ in the subscript for clarity.

To facilitate the analysis of overoptimization in RLHF without relying on expensive human feedback, Gao et al. (2023) introduced a setting in which a large RM, referred to as gold RM $R_{\text{gold}}$, is used as a replacement for the human feedback. By imitating a powerful $R_{\text{gold}}$ with a smaller RM, we can simulate the problem of imitating the very large "human RM" with a large proxy model as done when LLMs such as GPT-4 are trained (OpenAI et al., 2024).

## 3.2 Importance weighting

When training data $\{x^j, y^j\}_{j=1}^N \overset{\text{i.i.d.}}{\sim} P_0(X, Y)$ is available from a distribution with probability $P_0$, but we need to estimate the loss on a different distribution with probability $P_1$, we can use IW (Shimodaira, 2000). The expected loss $\mathcal{L}(\theta) = \mathbb{E}_{(x,y) \sim P_1}[l(x, y; \theta)]$ over $P_1$ with sample-wise loss $l$ can be written as

$$\mathbb{E}_{(x,y) \sim P_1(x,y)}[l(x, y; \theta)] = \mathbb{E}_{(x,y) \sim P_0(x,y)}\left[\frac{P_1(x,y)}{P_0(x,y)} l(x, y; \theta)\right].$$

$w(x,y) = P_1(x,y)/P_0(x,y)$ is called the importance weight. Importance weighted estimators usually have a higher variance, it is thus sometimes desirable to use variance-reduction techniques that incur some bias, such as flattening the importance weights with a hyperparameter $\eta \in [0, 1]$, i.e., $w^\eta(x,y) = (P_1(x,y)/P_0(x,y))^\eta$ (Shimodaira, 2000). Another related technique (Yamada et al., 2011) is to use the ratio $P_1(x,y)/P_0'(x,y)$ to the mixture $P_0'(x,y) = \alpha P_0(x,y) + (1 - \alpha)P_1(x,y)$ instead, with $\alpha \in [0, 1]$.

## 4 Distribution shift in RLHF

To motivate the need for our proposed method, we first discuss what happens in the normal RLHF setting as the policy changes.

The goal of RLHF is to obtain a policy $\pi_\phi$ that maximizes the expected return $J(\phi, \theta^{\text{H}})$ under an unknown human reward $R_{\theta^{\text{H}}}$. Therefore, at each optimization step $i$ we update the policy $\pi_\phi^i$ using estimates of the policy gradient $\nabla_\phi J(\phi, \theta)$, which depends on the learned RM parameters $\theta$. Since the true $\theta^{\text{H}}$ is unknown, we instead use the estimate $\hat{\theta}$ obtained by minimizing the sample average of $\mathcal{L}_{\text{RM}}^{\pi^1}$ (2) on the dataset $D_{\text{RM}}$ generated by the SFT policy $\pi^1$. As dataset size $N_{\text{RM}} \to \infty$, we know[1] that $\mathcal{L}_{\text{RM}}^{\pi^1}(\hat{\theta}) \to \mathcal{L}_{\text{RM}}^{\pi^1}(\theta^\circ)$, where $\theta^\circ = \arg\min_\theta \mathbb{E}_{(s,\mathbf{a}_w,\mathbf{a}_l) \sim P_{\pi^1}}[l_{\text{RM}}(s, \mathbf{a}_w, \mathbf{a}_l; \theta)]$ is the minimizer of the expected loss under $P_{\pi^1}$, i.e. the best approximation of the human RM $R_{\theta^{\text{H}}}$ on $P_{\pi^1}$, given unlimited data.

As training continues, $\pi^i$ shifts further from $\pi^1$ (Gao et al., 2023). In the policy updates, we are estimating the advantage of actions $\mathbf{a} \sim \pi^i$ and should thus use the $\theta$ that provides the closest approximation of human preferences on $P_{\pi^i}$, i.e., the minimizer $\theta^* = \arg\min_\theta \mathbb{E}_{(s,\mathbf{a}_w,\mathbf{a}_l) \sim P_{\pi^i}}[l_{\text{RM}}(s, \mathbf{a}_w, \mathbf{a}_l; \theta)]$. As $\theta^*$ is optimal by definition, we see that

$$\mathcal{L}_{\text{RM}}^{\pi^i}(\theta^\circ) \geq \mathcal{L}_{\text{RM}}^{\pi^i}(\theta^*),$$

---

[1]see Theorem 2 in Appendix A

where equality holds if $\pi^i = \pi^1$, i.e., typically only in the first training step. In the following training steps, $\hat{\theta}$ achieves a worse loss given unlimited training data and is an inconsistent estimator of $\theta^*$. Further, as we use $\hat{\theta}$ to estimate the policy gradient $\nabla_\phi J(\phi, \hat{\theta})$, the policy gradient estimate generally also becomes inconsistent. Likewise, the optimal policy parameters $\phi^\circ = \arg\max_\phi J(\phi, \theta^\circ)$ under $R_{\theta^\circ}$ generally do not match the optimal policy parameters $\phi^* = \arg\max_\phi J(\phi, \theta^*)$ under $R_{\theta^*}$ and therefore achieve a lower expected return

$$J(\phi^\circ, \theta^*) \leq J(\phi^*, \theta^*).$$

The whole RLHF process is therefore inconsistent in the sense that even with unlimited RM training data $D^{\mathrm{RM}}$ and unlimited RL interactions, it may not converge to the optimal policy $\pi_{\phi^*}$. Readers may notice that this is an interpretation of overoptimization as caused by covariate shift under model misspecification (Shimodaira, 2000).

## 5 Off-policy corrected reward modeling

As discussed above, using the estimate $\hat{\theta}$ of the RM parameters can lead to a suboptimal policy. We will thus explain how we can obtain a consistent estimate of $\theta^*$ in this section. One option is to sample new completions $(\mathbf{a}_0, \mathbf{a}_1) \sim \pi^i$ and obtain pairwise comparisons from human evaluators. However, as this is both costly and time-consuming, we instead propose to use IW as an attractive alternative that does not require any additional policy samples or labels.

As we have knowledge of the probability ratio $\pi^i(\mathbf{a} \mid s)/\pi^1(\mathbf{a} \mid s)$, we can rewrite the RM loss under the distribution of the policy $\pi^i$ as

$$\mathcal{L}_{\mathrm{RM}}^{\pi^i}(\theta) = \mathbb{E}_{(s, \mathbf{a}_\mathrm{w}, \mathbf{a}_\mathrm{l}) \sim P_{\pi^i}}[l_{\mathrm{RM}}(s, \mathbf{a}_\mathrm{w}, \mathbf{a}_\mathrm{l}; \theta)] = \mathbb{E}_{(s, \mathbf{a}_\mathrm{w}, \mathbf{a}_\mathrm{l}) \sim P_{\pi^1}}[w(s, \mathbf{a}_\mathrm{w}, \mathbf{a}_\mathrm{l})l_{\mathrm{RM}}(s, \mathbf{a}_\mathrm{w}, \mathbf{a}_\mathrm{l}; \theta)],\quad (4)$$

with importance weight $w(s, \mathbf{a}_\mathrm{w}, \mathbf{a}_\mathrm{l}) = P_{\pi^i}(s, \mathbf{a}_\mathrm{w}, \mathbf{a}_\mathrm{l})/P_{\pi^1}(s, \mathbf{a}_\mathrm{w}, \mathbf{a}_\mathrm{l})$ obtained by

$$w(s, \mathbf{a}_\mathrm{w}, \mathbf{a}_\mathrm{l}) = \frac{P(s)\pi^i(\mathbf{a}_\mathrm{w} \mid s)\pi^i(\mathbf{a}_\mathrm{l} \mid s)P(\mathbf{a}_\mathrm{w} > \mathbf{a}_\mathrm{l} \mid s)}{P(s)\pi^1(\mathbf{a}_\mathrm{w} \mid s)\pi^1(\mathbf{a}_\mathrm{l} \mid s)P(\mathbf{a}_\mathrm{w} > \mathbf{a}_\mathrm{l} \mid s)} = \frac{\pi^i(\mathbf{a}_\mathrm{w} \mid s)\pi^i(\mathbf{a}_\mathrm{l} \mid s)}{\pi^1(\mathbf{a}_\mathrm{w} \mid s)\pi^1(\mathbf{a}_\mathrm{l} \mid s)}. \quad (5)$$

Note that this is only possible if the support of $\pi^i$ is a subset of the support of $\pi^1$. Fortunately, for all LMs with softmax-outputs this is satisfied as $\pi(\mathbf{a}|s) > 0$ for all completions $\mathbf{a} \in \mathcal{A}$ and prompts $s \in \mathcal{S}$. Minimizing the sample average of (4) on $D_{\mathrm{RM}}$ gives us $\tilde{\theta}$, which we can subsequently use to estimate the policy gradient. We later show in Section 5.1 that the loss of $\mathcal{L}_{\mathrm{RM}}^{\pi^i}(\tilde{\theta}) \to \mathcal{L}_{\mathrm{RM}}^{\pi^i}(\theta^*)$ as we increase the sample size, matching the loss of the optimal $\theta^*$. As the policy changes with each update, we need to estimate $\tilde{\theta}$ newly after each update.

Readers familiar with the RL literature will recognize this as related to off-policy corrections (Sutton & Barto, 2018). We note that instead of accounting for the change of policy *in the policy gradient update*, we are here additionally accounting for it *in the reward modeling step*.

### 5.1 Estimation error analysis

We provide an estimation error bound (Mohri et al., 2018) for our off-policy corrected reward modeling approach. Due to space constraints we defer proofs and an estimation error bound for unweighted reward modeling to Appendix A. When evaluated on samples from $D_{\mathrm{RM}}$, $\hat{\theta} = \arg\min_\theta 1/N_{\mathrm{RM}} \sum_{j=1}^{N_{\mathrm{RM}}} l_{\mathrm{RM}}(s^j, \mathbf{a}_\mathrm{w}^j, \mathbf{a}_\mathrm{l}^j, \theta)$ is the empirical estimate of $\theta^\circ$ and $\tilde{\theta} = \arg\min_\theta 1/N_{\mathrm{RM}} \sum_{j=1}^{N_{\mathrm{RM}}} w(s^j, \mathbf{a}_\mathrm{w}^j, \mathbf{a}_\mathrm{l}^j)l_{\mathrm{RM}}(s^j, a_\mathrm{w}^j, \mathbf{a}_\mathrm{l}^j, \theta)$ is the empirical estimate of $\theta^*$.

**Theorem 1** (Estimation error bound for the proposed method). *Assume* $l_{\mathrm{RM}}(s, \mathbf{a}_\mathrm{w}, \mathbf{a}_\mathrm{l}; \theta) \leq C_l$ *and* $w(s, \mathbf{a}_\mathrm{w}, \mathbf{a}_\mathrm{l}) \in [0, W]$. *Define the margin function class*

$$\mathcal{F} = \left\{ f_\theta : (s, \mathbf{a}_\mathrm{w}, \mathbf{a}_\mathrm{l}) \mapsto R_\theta(s, \mathbf{a}_\mathrm{w}) - R_\theta(s, \mathbf{a}_\mathrm{l}) \ \middle|\ \theta \in \Theta \right\}.$$

*Then, with probability at least $1 - \delta$,*

$$\mathcal{L}_{\mathrm{RM}}^{\pi^i}(\tilde{\theta}) - \mathcal{L}_{\mathrm{RM}}^{\pi^i}(\theta^*) \le 4W\mathfrak{R}_{N_{\mathrm{RM}}}(\mathcal{F}) + WC_l\sqrt{\frac{2}{N_{\mathrm{RM}}}\ln\left(\frac{2}{\delta}\right)},$$

*where $\mathfrak{R}_{N_{\mathrm{RM}}}(\mathcal{F})$ is the Rademacher complexity of $\mathcal{F}$ at sample size $N_{\mathrm{RM}}$.*

This theorem implies that as the number of samples $N_{\mathrm{RM}}$ goes to infinity, $\mathcal{L}_{\mathrm{RM}}^{\pi^i}(\tilde{\theta})$ converges to $\mathcal{L}_{\mathrm{RM}}^{\pi^i}(\theta^*)$, because $\mathfrak{R}(\mathcal{F}) \to 0$ for parametric models with a bounded norm (Mohri et al., 2018). It is also intuitive that the upper bound tightens with a smaller $W$. The assumption of a bounded loss $l_{\mathrm{RM}} \le C_l$ is satisfied under a bounded reward.

It is also interesting to consider $\mathcal{L}_{\mathrm{RM}}^{\pi^i}(\hat{\theta}) - \mathcal{L}_{\mathrm{RM}}^{\pi^i}(\tilde{\theta})$. This quantifies how much worse it is to use standard reward modeling compared to the importance-weighted estimate, on finite samples. We can derive an upper bound in the following way:

$$\begin{aligned}
\mathcal{L}_{\mathrm{RM}}^{\pi^i}(\hat{\theta}) - \mathcal{L}_{\mathrm{RM}}^{\pi^i}(\tilde{\theta}) &\le \mathcal{L}_{\mathrm{RM}}^{\pi^i}\left(\hat{\theta}\right) - \mathcal{L}_{\mathrm{RM}}^{\pi^i}\left(\theta^*\right) \\
&= \mathcal{L}_{\mathrm{RM}}^{\pi^i}\left(\hat{\theta}\right) - \mathcal{L}_{\mathrm{RM}}^{\pi^1}\left(\hat{\theta}\right) + \mathcal{L}_{\mathrm{RM}}^{\pi^i}\left(\theta^*\right) - \mathcal{L}_{\mathrm{RM}}^{\pi^i}\left(\theta^*\right) + \mathcal{L}_{\mathrm{RM}}^{\pi^1}\left(\hat{\theta}\right) - \mathcal{L}_{\mathrm{RM}}^{\pi^1}\left(\theta^*\right) \\
&\le \underbrace{\mathcal{L}_{\mathrm{RM}}^{\pi^i}\left(\hat{\theta}\right) - \mathcal{L}_{\mathrm{RM}}^{\pi^1}\left(\hat{\theta}\right)}_{(a)} + \underbrace{\mathcal{L}_{\mathrm{RM}}^{\pi^1}\left(\theta^*\right) - \mathcal{L}_{\mathrm{RM}}^{\pi^i}\left(\theta^*\right)}_{(b)}.
\end{aligned}$$

When distributions $\pi^i$ and $\pi^1$ are not equivalent, (a) and (b) both become positive from the definition of the minimizers. These terms do not vanish even when $N_{\mathrm{RM}} \to \infty$. Hence, $\mathcal{L}_{\mathrm{RM}}(\hat{\theta}) - \mathcal{L}_{\mathrm{RM}}(\tilde{\theta})$ may be nonzero as $N_{\mathrm{RM}} \to \infty$, demonstrating the benefit of our method.

We note that, while we have shown that we can use IW to prevent the cause of inconsistency explained in Section 4, this does not mean that RLHF with off-policy corrected RMs will necessarily converge to an optimal policy. However, in the next section we will empirically show that it is useful in practice.

## 5.2 Practical implementation

As the policy changes after each policy update, we would need to also retrain the RM after each policy update, which is computationally infeasible. We thus implement an approximate method that only retrains the RM using IW after each $k$ policy updates. We repeat this $m$ times and denote the policy after iteration $i \in 1, 2, \ldots, m$ as $\pi^{i+1}$ and the RM trained with IW for the distribution of policy $P_{\pi^i}$ as $R_{\theta^i}$.

The main purpose of the KL-regularization is that it keeps the policy in a region where the RM is accurate, i.e., close to $\pi^1$. However, after using IW to obtain the RM $R_{\theta^i}$, $i > 1$, it is now more accurate close to the policy $\pi^i$. Thus, we also change the reference of the KL constraint to the previous policy $\pi^i$, that is, the policy $\pi^{i+1}$ is obtained by using PPO for $k$ updates with the reward $R_{\theta^i}(s, a) - D_{\mathrm{KL}}(\pi^{i+1}(\cdot \mid s) || \pi^i(\cdot \mid s))$, ensuring that the policy remains close to the distribution on which the RM is accurate.

We train all networks using AdamW (Loshchilov & Hutter, 2019), which maintains an estimate of the gradient moments. When switching to a new RM, the optimizer state as well as the previous value function are no longer useful and we thus reset both. As advantage estimates in PPO rely on the learned value function, using a small $k$ can therefore cause poor policy gradient estimates due to the value function being inaccurate in its initial training steps. In practice, we found that choosing a high $k$ such that the new policy is trained until convergence on the current RM performs well.

As mentioned above, importance weights can have a high variance. We thus use flattened, relative importance weights (Shimodaira, 2000; Yamada et al., 2011) with $\eta = 0.001$ and $\alpha = 0.9$ in all LM experiments. This reintroduces some bias but is advantageous with limited data (Shimodaira, 2000). We note that any other policy gradient methods such as RLOO (Ahmadian et al., 2024) or GRPO (Shao et al., 2024) can be used instead of PPO.

| Dataset | TL;DR Summarization | | | | Short Alpaca-Farm | |
|---|---|---|---|---|---|---|
| Model | Pythia 1B | Qwen 2.5 1.5B | P 2.8B | P 6.9B | P 1B | Q2.5 1.5B |
| SFT | $17.8 \pm 4.5\%$ | 31.6% | 28.1% | 27.5% | 12.5% | 39.3% |
| DPO | $43.7 \pm 0.7\%$ | 76.1% | 64.8% | 70.3% | 12.4% | 67.6% |
| WPO | $47.0 \pm 2.2\%$ | 77.1% | 64.5% | 71.5% | 10.9% | 64.5% |
| PPO/GRPO | $62.9 \pm 2.7\%$ | 80.5% | 47.3% | 64.9% | 21.3% | 66.1% |
| RLP-SPG | $57.1 \pm 9.7\%$ | 72.8% | - | - | 23.6% | 60.8% |
| Ours ($m = 2$) | $71.4 \pm 2.9\%$ | 87.7% | 63.4% | 78.6% | **25.3%** | 68.5% |
| Ours ($m = 3$) | **73.6±1.7%** | **89.6%** | **75.4%** | **79.8%** | 24.5% | **69.5%** |

Table 1: Win rate vs reference response, as judged by the gold RM. For Pythia 1B on TL;DR we run three different random seeds, corresponding to different SFT models and resulting RM datasets $D_{\mathrm{RM}}$, and report the mean and standard deviation. Other experiments use a single seed. We abbreviate Pythia as P and Qwen 2.5 as Q2.5. For the 6.9B model we use GRPO, all other experiments use PPO.

# 6 Experiments

As it is difficult to visualize LM experiments, we first illustrate our approach in a low-dimensional, didactic example and subsequently perform experiments with LM alignment.

## 6.1 Didactic experiment

We consider the case of a state space consisting of a single state $\mathcal{S} = \{s_0\}$ and a two-dimensional continuous action space $\mathcal{A} = [-1.5, 1.5]^2$. The reward function is a Rastrigin-like (Hoffmeister & Bäck, 1991) function and the initial policy $\pi^1$ is an isotropic, zero-mean normal distribution, as shown in Figure 1 (left). As RM we use a Multi-Layer-Perceptron (MLP) with one hidden layer with four units. We do not flatten the importance weights in this experiment. The results are shown in Figure 1.

The initial RM is accurate on the distribution of the initial policy $\pi^1$, but accuracy quickly degrades as the policy changes, leading the achieved true reward to worsen after initially improving. Accuracy here is defined as agreement between $R_\theta$ and $R_{\mathrm{gold}}$ on the ranking of pairs of samples drawn from $\pi$, i.e. $\mathbb{E}_{s \sim D_{\mathrm{RM}}, (a_0, a_1) \sim \pi(a|s)} \left[ I_{R_\theta(a_0) > R_\theta(a_1) = R_{\mathrm{gold}}(a_0) > R_{\mathrm{gold}}(a_1)} \right]$ with indicator function $I$. By retraining the RM using IW after 200,000 samples, we obtain a new RM $R_{\theta^2}$ that is more accurate in the proximity of the current policy $\pi^2$, which allows us to use it to train a better policy $\pi^3$. By repeating this process, we are able to converge to a close to optimal policy in three iterations, while standard PPO-RLHF stagnates.

## 6.2 Language modeling experiments

To evaluate our method when used to align LMs, we perform experiments with the gold-model setup (Gao et al., 2023) on the foundational summarization from human feedback task (Stiennon et al., 2020) and on a length-truncated version of the Alpaca-Farm chatbot dataset (Dubois et al., 2023). As gold RM $R_{\mathrm{gold}}$ we use the leading model on RewardBench (Lambert et al., 2024) at time of experiments under 10B parameters, specifically *Skywork-Reward-Llama-3.1-8B-v0.2* (Liu et al., 2024). We implement our method and baselines based on the code provided by Huang et al. (2024).[2] We report the win rate against the reference responses in each dataset as judged by the Gold RM. For our method, we retrain the RM after each $k = 100,000$ training samples. More implementation and experimental details can be found in Appendix D.

---

[2]Our implementation is shared at `github.com/JohannesAck/OffPolicyCorrectedRewardModeling`.

|  | Win Rate | Gold Score |
|---|---|---|
| PPO | 59.6% | -8.62 |
| PPO + Reset | 60.0% | -8.41 |
| PPO + New KL | 67.5% | -7.71 |
| Ours ($m = 2$) | 68.6% | -7.56 |
| Ours ($m = 3$) | 71.7% | -7.28 |

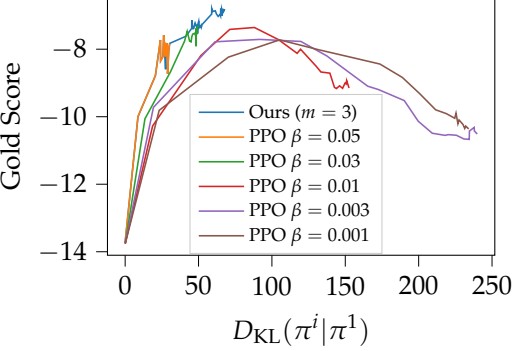

Table 2: Summarization with different ablations of our proposed method.

Figure 2: Our method and PPO with varied KL-regularization $\beta$.

### 6.2.1 Baseline comparison

As baselines, we compare our proposed approach with PPO (Stiennon et al., 2020), DPO (Rafailov et al., 2023), WPO (Zhou et al., 2024), and Reinforcement Learning on Policy Synthetic Preference Generation (RLP-SPG) (Lang et al., 2024). RLP-SPG samples outputs from the policy and adds synthetic labels, which are used to retrain the RM. For RLP-SPG we did not find a public implementation, we thus implemented it based on the paper.

We show our main results in Table 1. Our proposed method achieves a higher win rate than baselines, and improves upon PPO for both Pythia (Biderman et al., 2023) and Qwen 2.5 (Qwen et al., 2025) base-models, both in the chatbot and summarization tasks. In summarization, PPO is trained for 300,000 samples, RLP-SPG and our method with $m = 2$ are trained for 200,000 samples, our method with $m = 3$ is trained for 300,000 samples. We found DPO and WPO to perform worse with longer training and thus report their score when training for 100,000 samples in the summarization task. This finding is consistent with Xu et al. (2025, Appendix A.1) and we discuss it further in Appendix E. We also evaluate our method with 2.8B and 6.9B Pythia models, but only use selected baselines due to computational constraints.

### 6.2.2 Additional Experiments

To better understand our proposed method, we perform additional experiments with the Pythia-1B model on the summarization task, each with a single random seed.

**Method ablations** In our implementation we made two major design choices in addition to using the off-policy corrected RM: Firstly, we change the reference for the KL regularization from $D_{\mathrm{KL}}(\pi^i||\pi^1)$ to $D_{\mathrm{KL}}(\pi^i||\pi^{i-1})$. To test the effect of this change, we continue training the policy with the initial RM $R_{\theta 1}$ but new KL term $D_{\mathrm{KL}}(\pi^i||\pi^{i-1})$. We call this method "PPO + New KL". We also reset the value function parameters when changing the RM. Recent RL literature has explored parameter resets as a tool to increase plasticity, showing it to be beneficial in continuous control tasks (Nikishin et al., 2022; Dohare et al., 2024). We thus also provide a baseline, called "PPO + Reset", that resets the value function parameters and optimizer state but does not change the reward function or KL-term. As shown in Table 2, both ablations improve upon PPO, but do not reach the performance of our full method.

**KL-constraint strength** Changing the KL-term from $D_{\mathrm{KL}}(\pi^i||\pi^1)$ to $D_{\mathrm{KL}}(\pi^i||\pi^{i-1})$ after each iteration effectively weakens the KL regularization. We thus investigate whether simply weakening the strength $\beta$ of the KL-regularization in PPO can match the performance of our method. As shown in Figure 2, optimizing $\beta$ indeed improves the Gold score achieved by PPO, but it does not reach the performance of our method, both in terms of KL-Reward tradeoff and best achievable gold model score. Due to computational constraints, we did not optimize $\beta$ for our method and use the default $\beta = 0.05$ suggested by Huang et al. (2024).

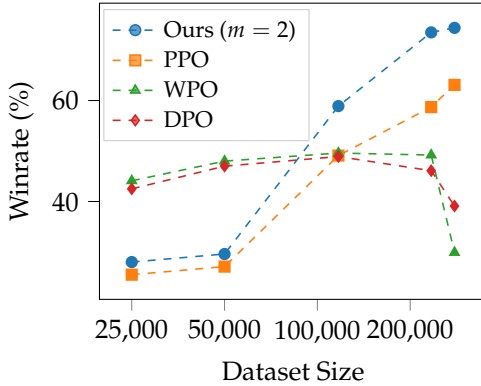

Figure 3: Training with different sizes of the preference-labeled dataset $D_{\text{RM}}$.

Figure 4: Varying the number of PPO updates $k$ before retraining the reward model.

| $\eta$ | $\alpha$ | Gold Score | Win Rate |
|--------|----------|------------|----------|
| 0.001 | 0.9 | $-6.83$ | 74.4% |
| 0.001 | 1.0 | $-6.88$ | 74.3% |
| 0.003 | 1.0 | $-7.36$ | 70.0% |
| 0.01 | 1.0 | $-6.87$ | 74.3% |
| 0.03 | 1.0 | $-7.12$ | 72.2% |
| PPO | - | $-7.92$ | 64.4% |

Table 3: Performance of Pythia-1B-deduped in the summarization task with different hyperparameters for flattened IW $\eta$ and relative IW $\alpha$, with $m = 2$.

| Method | GPT 4.1 Nano Win Rate |
|--------|-----------------------|
| SFT | 30.1% |
| PPO | 57.0% |
| Ours ($m = 2$) | 62.9% |
| Ours ($m = 3$) | 70.4% |

Table 4: Training and evaluation with feedback by GPT 4.1 nano instead of the gold reward model.

**Dataset size** IW can provide unbiased estimates but it increases the variance of the estimate (Shimodaira, 2000). We thus investigate how our method performs with less available data by running experiments with smaller datasets $D_{\text{RM}}$ and show the results in Figure 3. Even with less data the performance of our method degrades gracefully and still provides an improvement over PPO. Further, when few training data is available, the offline methods DPO and WPO perform better, while the online methods perform better with more data.

**Hyperparameters** We also evaluate the sensitivity of our approach to the newly introduced hyperparameters $k$, $\alpha$ and $\eta$. In Figure 4, we show that using a larger number of PPO updates $k$ before retraining the RM results in a significantly better KL-Reward trade-off and achieves a better final reward. We hypothesize that this is due to the value function being inaccurate during the early training stages, which we also visualize in the Appendix in Figure 7. In Table 3 we show the impact of the IW hyper-parameters $\eta$ and $\alpha$. While they are important, our method improves upon PPO with all tested values.

**GPT 4.1 Nano Experiments** In our other experiments we use a gold RM as proxy for human feedback. To validate our method with a more realistic synthetic setup, we also run experiments in which we label the training dataset $D_{\text{RM}}$ and evaluate the final policy with pairwise feedback provided by GPT 4.1 Nano. The results in Table 4 show that our method also performs well in this setting.

We provide training curves, further discussion of hyperparameters $k$ and $\eta$, runtime duration, results for longer DPO training, more training iterations $m = 5$, experiment details and output samples in Appendix E.

# 7 Conclusion

We have investigated the issue of overoptimization in reinforcement learning from human feedback as a type of distribution shift. This distribution shift occurs as the reward model (RM) is trained on the data distribution of the supervised fine-tuning policy $\pi^1$ but evaluated on the current policy $\pi^i$, which shifts further and further from $\pi^1$ during training. We have shown that estimating the RM parameters on actions from $\pi^1$ results in an inconsistent estimate of the RM parameters under the updated policy $\pi^i$ and consequently an inconsistent estimate of the policy gradient. By retraining the RM with importance weighting after each policy update we can address this issue. Empirically, we have shown that an approximate version which only retrains the RM after each $k$ updates is sufficient to significantly improve upon the performance of baseline methods on language model summarization and chatbot alignment tasks.

A limitation of our method is that it requires knowledge of the density of the SFT policy $\pi^1$ for all RM training samples. While this is not a problem if $D_{\text{RM}}$ is collected with a known SFT policy, it would be interesting to consider cases in which additional preference data from other policies is available.

## Acknowledgments

We thank Yivan Zhang and Thanawat Lodkaew for helpful discussions. TI was supported by KAKENHI Grant Number 22K17946. This work was supported by the Institute for AI and Beyond, UTokyo.

## Ethics statement

Our work provides a general-purpose method for RLHF and specifically focuses on an application to the alignment of LMs. Our work thus shares the same ethical concerns that apply to all preference-tuned LMs and their applications, but we do not foresee any additional, new ethical issues arising from our work.

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

# A Estimation error bounds

We first provide an estimation error bound for reward modeling without distribution shift and its proof. Next, we build on this result to provide an estimation error bound for our proposed importance weighted reward modeling method.

Note that for readability, the notation here differs from the main text slightly, as $\mathcal{L}$ always refers to $\mathcal{L}_{\text{RM}}$ and $n$ corresponds to $N_{\text{RM}}$. Also recall that $\mathcal{L}_{\text{RM}}$ refers to the population risk, while $R_\theta$ refers to a reward function, differing from the notation in Mohri et al. (2018).

## A.1 Reward modeling without distribution shift

We consider a parameterized reward function $R_\theta(s, \mathbf{a}) \in \mathbb{R}$, where $\theta \in \Theta$. We observe a training dataset

$$D = \{(s^i, \mathbf{a}_w^i, \mathbf{a}_l^i)\}_{i=1}^n \overset{\text{i.i.d.}}{\sim} p(s, \mathbf{a}_w, \mathbf{a}_l),$$

which consists of $n$ samples, each triple indicating that in prompt $s^i$, response $\mathbf{a}_w^i$ was preferred by a human labeler over response $\mathbf{a}_l^i$. In reward modeling under a Bradley–Terry or logistic-preference assumption, recall that the sample-wise loss is:

$$l_{\text{rm}}(s, \mathbf{a}_w, \mathbf{a}_l; \theta) = -\log\Big(\sigma\big(R_\theta(s, \mathbf{a}_w) - R_\theta(s, \mathbf{a}_l)\big)\Big), \quad \sigma(x) = \frac{1}{1 + e^{-x}}.$$

We previously define the population risk as:

$$\mathcal{L}(\theta) = \mathbb{E}_{p(s, \mathbf{a}_w, \mathbf{a}_l)}\Big[l_{\text{rm}}(s, \mathbf{a}_w, \mathbf{a}_l; \theta)\Big],$$

and the empirical risk on the dataset $D$ is

$$\widehat{\mathcal{L}}(\theta) = \frac{1}{n} \sum_{i=1}^n l_{\text{rm}}(s^i, \mathbf{a}_w^i, \mathbf{a}_l^i; \theta).$$

The empirical minimizer and true minimizer are

$$\hat{\theta} = \arg\min_{\theta \in \Theta} \widehat{\mathcal{L}}(\theta), \quad \theta^\circ = \arg\min_{\theta \in \Theta} \mathcal{L}(\theta),$$

respectively.

**Theorem 2** (Estimation error bound for the ordinary case)**.** *Assume there is a constant $C_l$ such that $l_{\text{rm}}(s, \mathbf{a}_w, \mathbf{a}_l; \theta) \leq C_l$ for all $\theta \in \Theta$ and all $(s, \mathbf{a}_w, \mathbf{a}_l)$. Define the margin function class*

$$\mathcal{F} = \Big\{ f_\theta : (s, \mathbf{a}_w, \mathbf{a}_l) \mapsto R_\theta(s, \mathbf{a}_w) - R_\theta(s, \mathbf{a}_l) \,\Big|\, \theta \in \Theta \Big\}.$$

*Then for any $\delta > 0$, with probability at least $1 - \delta$,*

$$\mathcal{L}(\hat{\theta}) - \mathcal{L}(\theta^\circ) \leq 4\mathfrak{R}_n(\mathcal{F}) + C_l \sqrt{\frac{2}{n} \ln\Big(\frac{2}{\delta}\Big)},$$

*where $\mathfrak{R}_n(\mathcal{F})$ is the Rademacher complexity of $\mathcal{F}$ at sample size $n$.*

This theorem implies that as the number of $n$ goes to infinity, $\mathcal{L}(\hat{\theta})$ converges to $\mathcal{L}(\theta^\circ)$ because $\mathfrak{R}(\mathcal{F}) \to 0$ for parametric models with a bounded norm (Mohri et al., 2018).

Note that we made an assumption $l_{\text{rm}} \leq C_l$. This is satisfied when the reward is bounded. If $R_\theta(s, a)$ remains in $[-B, B]$, then

$$f_\theta(s, \mathbf{a}_w, \mathbf{a}_l) = R_\theta(s, \mathbf{a}_w) - R_\theta(s, \mathbf{a}_l) \in [-2B, +2B].$$

The logistic loss $-\log\sigma(f_\theta)$ is then bounded above by $\max_{m \in [-2B, 2B]}[-\log\sigma(m)]$, which can serve as $C_l$.

*Proof.* We want an upper bound on $\sup_{\theta \in \Theta} \left| \widehat{\mathcal{L}}(\theta) - \mathcal{L}(\theta) \right|$. Since each loss is at most $C_l$, changing one sample $(s^i, \mathbf{a}_w^i, \mathbf{a}_l^i)$ can alter $\widehat{\mathcal{L}}(\theta)$ by at most $C_l / n$. By McDiarmid's inequality, for any $\epsilon > 0$,

$$\Pr\left[ \sup_{\theta \in \Theta}(\widehat{\mathcal{L}}(\theta) - \mathcal{L}(\theta)) - \mathbb{E}\left\{ \sup_{\theta \in \Theta}(\widehat{\mathcal{L}}(\theta) - \mathcal{L}(\theta)) \right\} \geq \epsilon \right] \leq \exp\left( -\frac{2\epsilon^2}{n \left( \frac{C_l^2}{n} \right)^2} \right) = \exp\left( -\frac{2n\epsilon^2}{C_l^2} \right).$$

Setting $\delta/2 = \exp\left( -\frac{2n\epsilon^2}{C_l^2} \right)$ yields $\epsilon = C_l \sqrt{\frac{1}{2n} \ln\left( \frac{2}{\delta} \right)}$. Thus, with probability at least $1 - \delta/2$,

$$\sup_{\theta \in \Theta}(\widehat{\mathcal{L}}(\theta) - \mathcal{L}(\theta)) \leq \mathbb{E}\left[ \sup_{\theta \in \Theta}(\widehat{\mathcal{L}}(\theta) - \mathcal{L}(\theta)) \right] + C_l \sqrt{\frac{1}{2n} \ln\left( \frac{2}{\delta} \right)}.$$

Next we handle the expectation term $\mathbb{E}[\sup_{\theta \in \Theta}(\widehat{\mathcal{L}}(\theta) - \mathcal{L}(\theta))]$. By a standard symmetrization argument (Vapnik, 1998), let $D$ be our $n$ i.i.d. samples and $D'$ be an independent ghost sample of the same size. Then

$$\mathbb{E}_D\left[ \sup_{\theta \in \Theta}(\widehat{\mathcal{L}}(\theta) - \mathcal{L}(\theta)) \right] = \mathbb{E}_D\left[ \sup_{\theta \in \Theta}(\widehat{\mathcal{L}}(\theta; D) - \mathbb{E}_{D'}[\widehat{\mathcal{L}}(\theta; D')]) \right] \leq \mathbb{E}_{D,D'}\left[ \sup_{\theta \in \Theta}(\widehat{\mathcal{L}}(\theta; D) - \widehat{\mathcal{L}}(\theta; D')) \right].$$

Note that

$$\widehat{\mathcal{L}}(\theta; D) = \frac{1}{n} \sum_{i=1}^{n} l_{\text{rm}}(x_i; \theta), \quad \widehat{\mathcal{L}}(\theta; D') = \frac{1}{n} \sum_{i=1}^{n} l_{\text{rm}}(x_i'; \theta),$$

where $x_i = (s^i, \mathbf{a}_w^i, \mathbf{a}_l^i)$ and $x_i' = (s'^i, \mathbf{a}_w'^i, \mathbf{a}_l'^i)$. Hence

$$\widehat{\mathcal{L}}(\theta; D) - \widehat{\mathcal{L}}(\theta; D') = \frac{1}{n} \sum_{i=1}^{n} \left[ l_{\text{rm}}(x_i; \theta) - l_{\text{rm}}(x_i'; \theta) \right].$$

Define

$$\Delta(\theta) = \sum_{i=1}^{n} \left[ l_{\text{rm}}(x_i; \theta) - l_{\text{rm}}(x_i'; \theta) \right].$$

Under $D, D'$, the pair $(x_i, x_i')$ has the same distribution as $(x_i', x_i)$, so $\Delta(\theta)$ and $-\Delta(\theta)$ share the same distribution. Introducing a Rademacher variable $\sigma_i \in \{\pm 1\}$, we obtain:

$$\mathbb{E}_{D,D'}\left[ \sup_{\theta \in \Theta} \Delta(\theta) \right] = \mathbb{E}_{D,D'}\left[ \sup_{\theta \in \Theta} \sum_{i=1}^{n} \left[ l_{\text{rm}}(x_i; \theta) - l_{\text{rm}}(x_i'; \theta) \right] \right]$$

$$\leq \mathbb{E}_{\sigma,D}\left[ \sup_{\theta} \sum_{i=1}^{n} \sigma_i l_{\text{rm}}(x_i; \theta) \right] + \mathbb{E}_{\sigma,D'}\left[ \sup_{\theta} \sum_{i=1}^{n} -\sigma_i l_{\text{rm}}(x_i'; \theta) \right].$$

But $-\sigma_i$ is again a Rademacher variable, so the second expectation equals the first:

$$\mathbb{E}_{D,D'}\left[ \sup_{\theta} \Delta(\theta) \right] \leq 2\mathbb{E}_{\sigma,D}\left[ \sup_{\theta} \sum_{i=1}^{n} \sigma_i l_{\text{rm}}(x_i; \theta) \right] = 2n\mathbb{E}_{\sigma,D}\left[ \sup_{\theta} \frac{1}{n} \sum_{i=1}^{n} \sigma_i l_{\text{rm}}(x_i; \theta) \right].$$

Hence

$$\mathbb{E}_D\left[ \sup_{\theta}(\widehat{R}(\theta) - \mathcal{L}(\theta)) \right] \leq 2\mathbb{E}_{\sigma,D}\left[ \sup_{\theta} \frac{1}{n} \sum_{i=1}^{n} \sigma_i l_{\text{rm}}(x_i; \theta) \right] = 2\mathfrak{R}_n\left( l_{\text{rm}} \circ \mathcal{F} \right),$$

where $l_{\text{rm}} \circ \mathcal{F}$ indicates the function class $\{(s, \mathbf{a}_w, \mathbf{a}_l) \mapsto l_{\text{rm}}(s, \mathbf{a}_w, \mathbf{a}_l; \theta) \mid \theta \in \Theta\}$. Combining these pieces, with probability at least $1 - \delta/2$,

$$\sup_{\theta \in \Theta}(\widehat{\mathcal{L}}(\theta) - \mathcal{L}(\theta)) \leq 2\mathfrak{R}_n\left( l_{\text{rm}} \circ \mathcal{F} \right) + C_l \sqrt{\frac{1}{2n} \ln\left( \frac{2}{\delta} \right)}.$$

A similar argument for the other direction, $\sup_{\theta \in \Theta}(\mathcal{L}(\theta) - \widehat{\mathcal{L}}(\theta))$, yields the same bound. Both hold simultaneously with probability at least $1 - \delta$, so

$$\sup_{\theta \in \Theta}|\widehat{\mathcal{L}}(\theta) - \mathcal{L}(\theta)| \leq 2\mathfrak{R}_n(l_{\mathrm{rm}} \circ \mathcal{F}) + C_l\sqrt{\frac{1}{2n}\ln\left(\frac{2}{\delta}\right)}.$$

Finally, by definition of $\hat{\theta}$ and $\theta^\circ$,

$$\mathcal{L}(\hat{\theta}) - \mathcal{L}(\theta^\circ) = \left(\mathcal{L}(\hat{\theta}) - \widehat{\mathcal{L}}(\hat{\theta})\right) + \left(\widehat{\mathcal{L}}(\hat{\theta}) - \widehat{\mathcal{L}}(\theta^\circ)\right) + \left(\widehat{\mathcal{L}}(\theta^\circ) - \mathcal{L}(\theta^\circ)\right)$$
$$\leq \sup_{\theta \in \Theta}\left(\mathcal{L}(\theta) - \widehat{\mathcal{L}}(\theta)\right) + 0 + \sup_{\theta \in \Theta}\left(\widehat{\mathcal{L}}(\theta) - \mathcal{L}(\theta)\right) \leq 2\sup_{\theta \in \Theta}|\widehat{\mathcal{L}}(\theta) - \mathcal{L}(\theta)|.$$

Thus with probability at least $1 - \delta$,

$$\mathcal{L}(\hat{\theta}) - \mathcal{L}(\theta^\circ) \leq 2\left[2\mathfrak{R}_n(l_{\mathrm{rm}} \circ \mathcal{F}) + C_l\sqrt{\frac{1}{2n}\ln\left(\frac{2}{\delta}\right)}\right] = 4\mathfrak{R}_n(l_{\mathrm{rm}} \circ \mathcal{F}) + 2C_l\sqrt{\frac{1}{2n}\ln\left(\frac{2}{\delta}\right)}.$$

Since $l_{\mathrm{rm}} \circ \mathcal{F}$ is 1-Lipschitz[3] in $f_\theta$, $\mathfrak{R}_n(l_{\mathrm{rm}} \circ \mathcal{F}) \leq \mathfrak{R}_n(\mathcal{F})$ based on Talagrand's lemma. Therefore,

$$\mathcal{L}(\hat{\theta}) - \mathcal{L}(\theta^\circ) \leq 4\mathfrak{R}_n(\mathcal{F}) + C_l\sqrt{\frac{2}{n}\ln\left(\frac{2}{\delta}\right)},$$

which completes the proof. $\qquad\square$

### A.2 Importance-weighted reward modeling

The responses may come from a policy $\pi$ but we wish to minimize the risk under a different policy $\pi'$. Let $q(s, \mathbf{a}_{\mathrm{w}}, \mathbf{a}_{\mathrm{l}})$ be the joint distribution that corresponds to $\pi$ and $p(s, \mathbf{a}_{\mathrm{w}}, \mathbf{a}_{\mathrm{l}})$ be the joint distribution that corresponds to $\pi'$. Note that this $p$ is the same $p$ that we used to define $\mathcal{L}(\theta)$ earlier. We can rewrite the previous population risk as

$$\mathcal{L}'(\theta) = \mathbb{E}_{q(s, \mathbf{a}_{\mathrm{w}}, \mathbf{a}_{\mathrm{l}})}\left[w(s, \mathbf{a}_{\mathrm{w}}, \mathbf{a}_{\mathrm{l}})l_{\mathrm{rm}}(s, \mathbf{a}_{\mathrm{w}}, \mathbf{a}_{\mathrm{l}}; \theta)\right] = \mathcal{L}(\theta), \quad w(s, \mathbf{a}_{\mathrm{w}}, \mathbf{a}_{\mathrm{l}}) = \frac{\pi'(\mathbf{a}_{\mathrm{w}} \mid s)\pi'(\mathbf{a}_{\mathrm{l}} \mid s)}{\pi(\mathbf{a}_{\mathrm{w}} \mid s)\pi(\mathbf{a}_{\mathrm{l}} \mid s)}.$$

If our sample size $n$ is i.i.d. from $q$, we define the empirical estimator as

$$\widehat{\mathcal{L}}'(\theta) = \frac{1}{n}\sum_{i=1}^{n} w(s^i, \mathbf{a}_{\mathrm{w}}^i, \mathbf{a}_{\mathrm{l}}^i)l_{\mathrm{rm}}(s^i, \mathbf{a}_{\mathrm{w}}^i, \mathbf{a}_{\mathrm{l}}^i; \theta).$$

Let $\tilde{\theta} = \arg\min_\theta \widehat{\mathcal{L}}'(\theta)$ and $\theta'_* = \arg\min_\theta \mathcal{L}'(\theta)$. Since $\mathcal{L}'(\theta) = \mathcal{L}(\theta)$, the minimizers are also the same, i.e., $\theta'_* = \theta^*$.

**Theorem 1** (Estimation error bound for the proposed method). *Assume $l_{\mathrm{rm}}(s, \mathbf{a}_{\mathrm{w}}, \mathbf{a}_{\mathrm{l}}; \theta) \leq C_l$ and $w(s, \mathbf{a}_{\mathrm{w}}, \mathbf{a}_{\mathrm{l}}) \in [0, W]$. Then, with probability at least $1 - \delta$,*

$$\mathcal{L}(\tilde{\theta}) - \mathcal{L}(\theta^*) \leq 4W\mathfrak{R}_n(\mathcal{F}) + WC_l\sqrt{\frac{2}{n}\ln\left(\frac{2}{\delta}\right)},$$

*where $\mathcal{F} = \{f_\theta\}$ is the same margin-function class as before.*

This theorem also demonstrates that when $n$ goes to infinity, $\mathcal{L}(\tilde{\theta})$ will converge to $\mathcal{L}(\theta^*)$. It is also intuitive that the upper bound becomes tigher with a smaller $W$.

---

[3]Let $f_\theta(s, \mathbf{a}_{\mathrm{w}}, \mathbf{a}_{\mathrm{l}}) = R_\theta(s, \mathbf{a}_{\mathrm{w}}) - R_\theta(s, \mathbf{a}_{\mathrm{l}})$. Sample-wise loss is $l_{\mathrm{rm}}(s, \mathbf{a}_{\mathrm{w}}, \mathbf{a}_{\mathrm{l}}; \theta) = -\log\left(\sigma(f_\theta(s, \mathbf{a}_{\mathrm{w}}, \mathbf{a}_{\mathrm{l}}))\right)$. A derivative check shows $\frac{d}{dm}\left[-\log\sigma(m)\right] = \sigma(m) - 1$, which takes values in $[-1, 0]$. Hence $l_{\mathrm{rm}}(\cdot; \theta)$ is 1-Lipschitz with respect to the margin $f_\theta$.

*Proof.* In the importance-weighted case, we simply consider the new sample-wise loss function

$$l_{\text{IS}}(x; \theta) = w(x) l_{\text{rm}}(x; \theta),$$

where $x = (s, \mathbf{a}_{\text{w}}, \mathbf{a}_{\text{l}})$ and $w(\cdot) \in [0, W]$. Then the same argument as Theorem 2 applies except for the following two changes:

$$l_{\text{IS}}(x; \theta) \leq WC_l, \qquad l_{\text{IS}}(\cdot; \theta) \text{ is } (W \times 1)\text{-Lipschitz in the margin,}$$

because $l_{\text{rm}}$ is 1-Lipschitz and we multiply by at most $W$. Hence all factors in the uniform deviation bound gain a factor $W$. We get

$$\sup_{\theta \in \Theta} \left| \widehat{\mathcal{L}}'(\theta) - \mathcal{L}'(\theta) \right| \leq 2\mathfrak{R}_n \left( l_{\text{IS}} \circ \mathcal{F} \right) + WC_l \sqrt{\frac{1}{2n} \ln\left(\frac{2}{\delta}\right)},$$

and $l_{\text{IS}}(\cdot; \theta)$ is $W$-Lipschitz in $f_\theta$, so $\mathfrak{R}_n(l_{\text{IS}} \circ \mathcal{F}) \leq W\mathfrak{R}_n(\mathcal{F})$. Finally,

$$\mathcal{L}'(\tilde{\theta}) - \mathcal{L}'(\theta'_*) \leq 2 \sup_{\theta \in \Theta} \left| \widehat{\mathcal{L}}'(\theta) - \mathcal{L}'(\theta) \right| \leq 4W\mathfrak{R}_n(\mathcal{F}) + WC_l \sqrt{\frac{2}{n} \ln\left(\frac{2}{\delta}\right)}.$$

Since $\mathcal{L}'(\theta) = \mathcal{L}(\theta)$ and $\theta'_* = \theta^*$

$$\mathcal{L}(\tilde{\theta}) - \mathcal{L}(\theta^*) \leq 4W\mathfrak{R}_n(\mathcal{F}) + WC_l \sqrt{\frac{2}{n} \ln\left(\frac{2}{\delta}\right)}.$$

This completes the proof. $\qquad\qquad\qquad\qquad\qquad\qquad\qquad\qquad\qquad\qquad\qquad\square$

## B  Comparison to Weighted Preference Optimization

Weighted Preference Optimization (WPO) (Zhou et al., 2024) is an alignment method that builds upon Direct Preference Optimization (DPO) (Rafailov et al., 2023) by adding a weighting term that imitates online learning.

**DPO**  aims to to solve the same RLHF problem setting as the RLHF formulation proposed by Stiennon et al. (2020) and used in our work: We want to obtain a policy $\pi^i$, that maximizes the regularized reward

$$\max_{\pi^i} \mathbb{E}_{\mathbf{a} \sim \pi^i} \left[ \hat{r}(s, \mathbf{a}) - \beta D_{\text{KL}}(\pi^i(\cdot \mid s) || \pi^1(\cdot \mid s)) \right],$$

where $\hat{r}(s, a)$ is again a reward following the BT-model as in (1). However, DPO does this without training a reward model and without choosing new actions with the policy. Instead, they derive a maximum likelihood formulation that minimizes the following objective:

$$\mathcal{L}_{\text{DPO}}(\pi^i; \pi^1) = -\mathbb{E}_{(s,\mathbf{a}_{\text{w}},\mathbf{a}_{\text{l}}) \sim D^{\text{RM}}} \left[ \log \sigma(\beta \log \frac{\pi^i(\mathbf{a}_{\text{w}} \mid s)}{\pi^1(\mathbf{a}_{\text{w}} \mid s)} - \beta \log \frac{\pi^i(\mathbf{a}_{\text{l}} \mid s)}{\pi^1(\mathbf{a}_{\text{l}} \mid s)}) \right],$$

and we define $l_{\text{DPO}}(s, \mathbf{a}_{\text{w}}, \mathbf{a}_{\text{l}}) = \log \sigma(\beta \log \frac{\pi^i(\mathbf{a}_{\text{w}}|s)}{\pi^1(\mathbf{a}_{\text{w}}|s)} - \beta \log \frac{\pi^i(\mathbf{a}_{\text{l}}|s)}{\pi^1(\mathbf{a}_{\text{l}}|s)})$.

The advantage of this DPO loss is that we do not need to generate new actions $a$ and can thus significantly speed up training. A disadvantage is that actions $a$ are no longer generated by the current policy $\pi^i$, thus it is an off-policy method.

**WPO**  Zhou et al. (2024) now correctly point out that due to DPO being off-policy, the action distribution produced by the policy no longer matches the preference dataset and address this issue by "simulating" on-policy RL. To do so they propose a weighted version of the DPO loss function:

$$\mathcal{L}_{\text{WPO}}(\pi^i; \pi^1) = -\mathbb{E}_{(s,\mathbf{a}_{\text{w}},\mathbf{a}_{\text{l}}) \sim D^{\text{RM}}} \left[ w(s, \mathbf{a}_{\text{w}}) w(s, \mathbf{a}_{\text{l}}) l_{\text{DPO}}(s, \mathbf{a}_{\text{w}}, \mathbf{a}_{\text{l}}) \right],$$

with weight $w(s, \mathbf{a})$. As weight they propose to use the policy probability $w(s, \mathbf{a}) = \pi^i(\mathbf{a} \mid s)$.[4] This is a reasonable heuristic choice, giving more weight to the actions more likely under the current policy.

However, as we have shown in Section 5, to obtain a proper off-policy correction, we need to use the probability ratio $\frac{\pi^i(\mathbf{a}|s)}{\pi^1(\mathbf{a}|s)}$. From this we can see that while WPO intends to optimize the loss

$$\mathbb{E}_{(\mathbf{a}_{\mathrm{w}},\mathbf{a}_{\mathrm{l}}) \sim \pi^i(a|s)} \left[ l_{\mathrm{DPO}}(s, \mathbf{a}_{\mathrm{w}}, \mathbf{a}_{\mathrm{l}}) \right] = \mathbb{E}_{(\mathbf{a}_{\mathrm{w}},\mathbf{a}_{\mathrm{l}}) \sim \pi^1(a|s)} \left[ \frac{\pi^i(\mathbf{a}_{\mathrm{w}} \mid s)}{\pi^1(\mathbf{a}_{\mathrm{w}} \mid s)} \frac{\pi^i(\mathbf{a}_{\mathrm{l}} \mid s)}{\pi^1(\mathbf{a}_{\mathrm{l}} \mid s)} l_{\mathrm{DPO}}(s, \mathbf{a}_{\mathrm{w}}, \mathbf{a}_{\mathrm{l}}) \right]$$

they are instead optimizing the loss

$$\mathbb{E}_{(\mathbf{a}_{\mathrm{w}},\mathbf{a}_{\mathrm{l}}) \sim \pi^1(a|s)} \left[ \pi^i(\mathbf{a}_{\mathrm{w}} \mid s) \pi^i(\mathbf{a}_{\mathrm{l}} \mid s) l_{\mathrm{DPO}}(s, \mathbf{a}_{\mathrm{w}}, \mathbf{a}_{\mathrm{l}}) \right] \neq \mathbb{E}_{(\mathbf{a}_{\mathrm{w}},\mathbf{a}_{\mathrm{l}}) \sim \pi^i(a|s)} \left[ l_{\mathrm{DPO}}(s, \mathbf{a}_{\mathrm{w}}, \mathbf{a}_{\mathrm{l}}) \right]],.$$

This is only a correct off-policy correction (up to scaling by a different constant) under the assumption of a uniform SFT policy $\pi^1(\mathbf{a} \mid s) \propto c$ with some constant c. This is not the case for most practical SFT policies, but it may be a reasonable assumption if we do not have access to the policy $\pi^1$ under which the dataset was generated. We note that in practice we usually have access to $\pi^1$.

We would also like to highlight that despite not providing a correct off-policy correction, WPO nonetheless provides a consistent improvement on the performance of DPO in our experiments.

## C  Additional related work

We provide an extended discussion of related work here.

**Better Reward Models**  Coste et al. (2024); Zhang et al. (2024); Eisenstein et al. (2024) proposed different ways of training and using ensembles of RMs and show that training with them results in a better alignment performance. Ramé et al. (2024) proposed to use a weight averaged combination of multiple RMs. These papers address inaccuracy of the RM, but do not address distribution shift directly and could thus be combined with our method.

**Iterated RLHF**  LLAMA 3 (Dubey et al., 2024) used an iterative alignment method which involves multiple stages of rejection sampling, SFT and DPO. Dong et al. (2024) presented RLHF Workflow, an iterative DPO method. Both involve doing multiple rounds of post-training including DPO, sampling from the model and labeling pairs of outputs with a pretrained RM. They both aimed to bridge the gap between DPO and PPO by sampling online data from the policy, while we are instead trying to improve PPO by correcting for the distribution shift without new samples or labels. Concurrently to our work, ProRL (Liu et al., 2025) proposes a method called "Reference Resets", which is identical to our baseline "PPO+New KL" in Table 2.

**Theoretical analyses of distribution shift in RLHF**  Song et al. (2024) provided a theoretical analysis comparing DPO and PPO under different support assumptions for $\pi^i$ and $\pi^1$. They proposed a method that is effective in bringing the performance of DPO closer to PPO, while our method focuses on improving PPO by improving the RM. Fluri et al. (2025) provided a theoretical analysis of RLHF, showing that under distribution shift an RM with small training error can lead to a policy with a large regret. This is similar to the motivation of our work and related to our discussion in Section 4, but we propose a method that addresses this issue starting from the cause of the RM error, while they focused on a theoretical analysis of the regret under an RM with known bounded error. Li et al. (2024) provide a sample complexity analysis of off-policy evaluation in RLHF, when using a Multi-Layer perceptron as RM. They also consider the effect of distribution shift in the reward modeling step and provide an estimation error bound that accounts for it.

---

[4]They also propose to use three other scaled variants of the policy probability, however they all have the same fundamental issue.

# D Experiment details

---

**Algorithm 1** PPO with OCRM

---

**Input:** Pretrained LM $\pi^{\mathrm{PT}}$, SFT dataset $D_{\mathrm{SFT}}$, iterations $m$, updates per iteration $k$

$\pi^1 \leftarrow \arg\min_\pi \mathbb{E}_{(s,\mathbf{a}) \sim D_{\mathrm{SFT}}}[l_{\mathrm{CE}}(s, \mathbf{a}, \pi(s))]$

Sample $(a_0, a_1) \sim \pi^1(a|s)$, label with preferences to obtain $D_{\mathrm{RM}} = \{(s^j, \mathbf{a}_{\mathrm{w}}^j, \mathbf{a}_{\mathrm{l}}^j)\}_{j=1}^{N_{\mathrm{RM}}}$

**for** $i = 1$ **to** $m$ **do**

    $R_{\theta i} \leftarrow \arg\min_\theta \mathbb{E}_{D_{\mathrm{RM}}}[w(s, \mathbf{a}_{\mathrm{w}}, \mathbf{a}_{\mathrm{l}})l_{\mathrm{RM}}(s, \mathbf{a}_{\mathrm{w}}, \mathbf{a}_{\mathrm{l}}, \theta)]$ with $w(s, \mathbf{a}_{\mathrm{w}}, \mathbf{a}_{\mathrm{l}}) = \frac{\pi^i(\mathbf{a}_1|s)\pi^i(\mathbf{a}_2|s)}{\pi^1(\mathbf{a}_1|s)\pi^1(\mathbf{a}_2|s)}$

    $\pi^{i+1} \leftarrow k$ PPO training steps with reward $R_{\theta,i+1}(s, \mathbf{a}) - D_{\mathrm{KL}}(\pi_\phi(\cdot \mid s)||\pi^i(\cdot \mid s))$

    Reset value network parameters and optimizer state for PPO

**end for**

---

## D.1 Didactic experiment

To illustrate our approach in an easily visualizable domain, we use a stateless environment with two-dimensional action space $\mathcal{A} = [-1.5, 1.5]^2$. The reward function is a Rastrigin-like function, specifically $r(s, a) = ||a|_2 - 0.7| + \sin(4\pi a^0) + \sin(6\pi a^1)$, where $a^1$ and $a^2$ are the first and second element of the action. The initial policy $\pi^1$ (corresponding to the SFT policy) is a zero-mean Gaussian with diagonal covariance matrix $\Sigma = 0.7I_2$. The reward model is an MLP with a single hidden layer with four units and tanh activations. We train the RM for 50 epochs on the $D_{\mathrm{RM}}$ and then use it to train a policy with PPO. The policy and value network each consist of MLPs with two hidden layers, 64 units and ReLU activations. After each 200,000 samples we retrain the RM with importance weighting. Here we use the raw importance weights without flattening or relative importance weighting.

## D.2 Language model experiments

*Training Samples* in the figures refers to samples seen (DPO/WPO) or generated (PPO, RLP-SPG, Ours) by the language model. For evaluation, we use sampling temperature 0.01.

All SFT models are trained on the train split for one epoch. The SFT models are then used to generate 278,496 training pairs $(s, \mathbf{a}_0, \mathbf{a}_1)$, with $s$ sampled from the SFT dataset and with sampling temperature 1.0. These pairs are then evaluated with the *Skywork-Reward-Llama-3.1-8B-v0.2* model (Liu et al., 2024). Note that we do not sample labels based on the probability implied by the Bradley-Terry model (1), but instead always choose the higher reward as winner, as done by Gao et al. (2023).

## D.3 Datasets

For the summarization experiments we use the TL;DR dataset as provided by Huang et al. (2024). We report the gold-score, win rate, and KL-divergence on the validation split of the summarization dataset provided by Huang et al. (2024). Huang et al. (2024) also reported their win rates on the validation split.

To test the performance of our method when used to train a chatbot, we use a modified version of the Alpaca-Instructions (Dubois et al., 2023) dataset. The Alpaca-Instructions dataset mostly contains short requests typical for a chatbot, but also a few long examples with around 1000 tokens. These require a larger context size for the language model and would significantly increase the training cost, we thus remove them from the training and validation datasets. Further, Alpaca-Instructions contains separate data splits intended for SFT, reward modeling and PPO. We combine them to a single training dataset used to train the SFT model and then sample the $D_{\mathrm{RM}}$ dataset from the SFT model.

With the dataset now being truncated to queries of at most 512 tokens and outputs of at most 106 tokens, we can use the same codebase as we used for the summarization experiments

in these experiments as well. We report the gold win rate on the length-filtered validation split of Alpaca-Farm, not Alpaca-Eval. The code used to filter the dataset and the resulting dataset are available in our GitHub repository.

### D.3.1 PPO and proposed method

We base the implementation of our method and PPO on the implementation provided by Huang et al. (2024) and keep their architectural and implementation details, as well as hyperparameters, except where otherwise noted. Notably, following Stiennon et al. (2020), in the implementation of Coste et al. (2024) the RM is initialized with the parameters of the SFT model and thus has the same parameter count. Rollouts are generated with sampling temperature 0.7.

Due to computational constraints, we were not able to perform an extensive hyper-parameter optimization for Pythia 1B. We thus use $\beta = 0.05$ in all Pythia 1B and Pythia 2.8B experiments, except for the ablation of it in Figure 2. This evaluation showed that $\beta = 0.03$ achieves a higher gold score than $\beta = 0.05$, but as it was run after finishing the main experiments and we did not tune $\beta$ for our method, we continued to use $\beta = 0.05$ for PPO in all experiments.

For Qwen 2.5 1.5B we increased the KL constraint strength to $\beta = 0.1$. For Pythia 6.9B we used GRPO instead of PPO-RLHF and use $\beta = 0.1$. Note that the values for $\beta$ in GRPO and PPO-RLHF are not directly comparable, as PPO-RLHF treats the KL constraint as part of the reward and applies it token-wise, while GRPO treats it as a separate loss term. Further, for the Pythia 6.9B experiment, we use an RM initialized with a Pythia 1B SFT model to conserve VRAM.

We show in Table 6 that increasing the iterations to $m = 5$ increases performance futher, however, we did not have the resources to evaluate this for all models/datasets.

### D.3.2 RLP-SPG

We could not find a public implementation of RLP (Lang et al., 2024), RLP-UML or RLP-SPG, we thus re-implemented RLP-SPG as it was the better performing method of the two in their experiments. Given the RL model $\pi^2$ we thus generate 10 replies for each prompt $s \in D_{RM}$. We then use *deberta-xlarge-mnli* (He et al., 2021) to calculate entailment scores for each pair of replies and divide the replies into clusters of replies such that all responses in a cluster mutually entail each other. If one cluster meets the threshold size of at least four replies, we sample $\mathbf{a}_w$ from the largest cluster and $\mathbf{a}_l$ from any other cluster. Lang et al. (2024) use a threshold of five replies and used the *deberta-large* model, however, we found a threshold four replies and the larger *deberta-xlarge* to perform slightly better.

### D.3.3 DPO and WPO

For DPO we are using the implementation provided by Coste et al. (2024), from the same suite of implementation as the PPO implementation. We modify it to include the weighting term for our WPO experiments. For the KL-regularization strength in DPO and WPO we evaluated the values $\beta \in \{0.3, 0.1, 0.03, 0.01\}$ on the Pythia 1B summarization task and found $\beta = 0.1$ to provide the best performance at the end of training and we thus used $\beta = 0.1$ for both DPO and WPO. For WPO we implemented the length-normalized weighting method, as it is the only method that was shared in the publicly available implementation (Zhou et al., 2024).

## E Additional results

### E.1 Reward Curves

In Figure 5 we show the evolution of the reward of our method and baselines during training, as well as the KL-Reward tradeoff. We can see that our method performs the same as PPO until the first retraining of the RM at 100,000 samples and then improves significantly, while

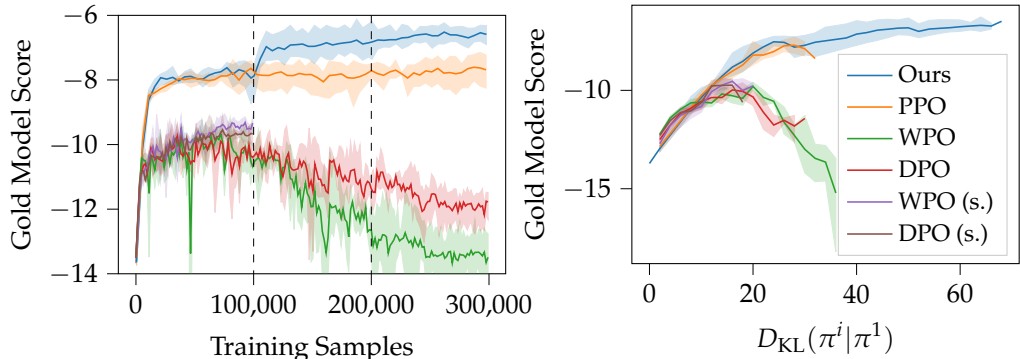

Figure 5: Comparison with baselines across three random seeds each with 95% confidence intervals on the summarization task with Pythia-1B. We train DPO (s.)(short) and WPO (s.) for 100,000 samples as they degrade with longer training. (see also Appendix E.8). The dashed lines indicate retraining of the RM in our proposed approach.

PPO stagnates. To be able to bootstrap confidence intervals for the KL-$R_{\text{gold}}$ plot in Figure 5 (right), we first interpolate the KL-$R_{\text{gold}}$ curve for each run to values on a fixed grid of KL values by linear interpolation. We then bootstrap the CIs for each KL value. The CIs for the samples-score plot are likewise boot-strapped.

## E.2 Importance weight flattening hyperparameters

| $\eta$ | $\alpha$ | $R_{\text{gold}}$ | Win Rate |
|---|---|---|---|
| 0.001 | 0.9 | $-6.83$ | 74.4% |
| 0.001 | 1.0 | $-6.88$ | 74.3% |
| 0.003 | 1.0 | $-7.36$ | 70.0% |
| 0.01 | 1.0 | $-6.87$ | 74.3% |
| 0.03 | 1.0 | $-7.12$ | 72.2% |
| PPO | - | $-7.92$ | 64.4% |

Table 5: Performance of Pythia-1B-dedup in the summarization task with different hyperparameters for flattened IW $\eta$ and relative IW $\alpha$. Here $m = 2$

Recall that we use flattened, relative importance weights $w_\alpha^\eta(x, y) = \left( \frac{P_1(x,y)}{\alpha P_0(x,y) + (1-\alpha) P_1(x,y)} \right)^\eta$ (Shimodaira, 2000; Yamada et al., 2011). We used $\eta = 0.001$ and $\alpha = 0.9$ in all experiments.

We note that using both flattened and relative importance weights can be considered redundant, as relative IW was proposed as an alternative to flattened IW (Yamada et al., 2011). As shown in Table 5, using relative IW with $\alpha = 0.9$ does not result in a noticeable difference compared to using only $\alpha = 1.0$, i.e. only using flattened importance weights. Due to limited computational resources and the small effect, we did not rerun all experiments without relative IW. We further note that in our alpaca experiments each action has 106 tokens, such that $\eta = 0.001$ and $\alpha = 1.0$ approximately uses the geometric mean of the token-wise density ratios. We believe that this can be a good guideline in practice.

We also ran additional experiments to investigate the effect of different values for $\eta$. The choice of the hyperparameter $\eta$, indeed has a noticeable effect on the performance of our proposed method, but all tested values result in an improvement over PPO.

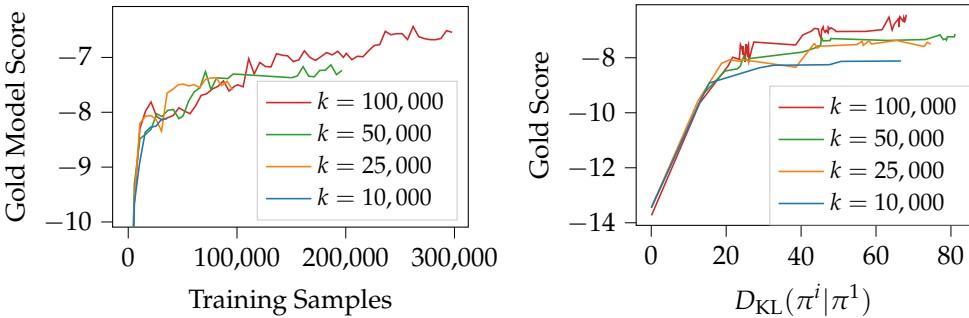

Figure 6: Evaluation of our approach with different values for $k$.

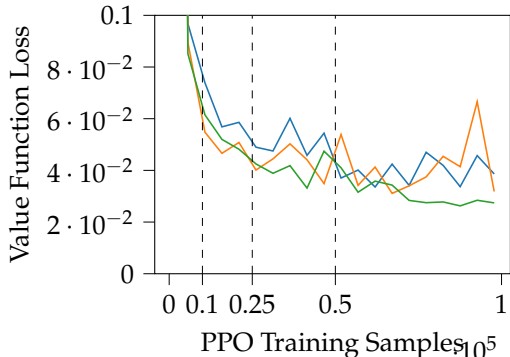

Figure 7: Value loss during PPO training with $k = 100,000$ for different random seeds. Note that during the early training stages the value function is still inaccurate, which might contribute to larger values for $k$ performing better.

### E.3  $k$ hyperparameter sensitivity

We also investigated how the number of policy updates $k$ performed before retraining the RM affects the performance of our approach and ran additional experiments with Pythia 1B on the summarization task. The results in Figure 6 show that a higher $k$ is advantageous in the asymptotic gold model score and also achieves a better gold model score for the same $D_{\mathrm{KL}}$. One possible explanation is that the value function is inaccurate in the first few updates after training a new reward model, leading to inaccurate policy updates. We thus also show the value loss when using $k = 100,000$ with three different random seeds in Figure 7. We can see that especially when training for less than $20,000$ samples the value loss is high, potentially explaining why $k = 10,000$ performs significantly worse.

However, when we are constrained by the number of PPO updates we can perform, a smaller $k$ could be beneficial as visible in the beginning of Figure 6.

### E.4  Longer Training

| Method | PPO | Ours ($m = 2$) | Ours ($m = 3$) | Ours ($m = 4$) | Ours ($m = 5$) |
|---|---|---|---|---|---|
| Win Rate | 59.7% | 68.6% | 71.7% | 74.4% | 74.9% |

Table 6: More iterations $m = 5$ for Pythia 1B on the TL;DR task with a single random seed.

In our main experiments, we restricted our approach to $m = 3$ iterations due to computational constraints. However, as there is no saturation being visible yet, we perform an additional experiment with $m = 5$ on a single seed with Pythia 1B on the TL;DR task. The

results in Table 6 show a continued increase in gold model win rate for $m = 4$ and $m = 5$, however it seems to saturate for $m = 5$.

### E.5 GPT 4.1 Nano Feedback

In our other experiments we use a Gold model, specifically *Skywork-Reward-Llama-3.1-8B-v0.2*, as proxy for human feedback, both to generate the training data and for evaluation. To alleviate concerns about this problem setup, we additionally train and evaluate a Pythia 1B model with feedback from GPT 4.1 nano instead. We use the following prompt format:

*You will be shown a Reddit post and two summaries. You will be asked to rate which one captures the content of the post better.*

<POST> {post} </POST>

<SUMMARY A> {response_a} </SUMMARY A>

<SUMMARY B> {response_b} </SUMMARY B>

*Please reply only with the letter "A" or "B", for which summary you think is better. Pick one, even if neither is good!*

We experimentally found a slight bias towards the first response. For evaluation, we thus input each pair of responses in both orders and count it as a tie if the preferences are not consistent. The win rate is shown in Table 7 and shows a significant improvement, consistent with the results of our gold model experiments.

| Method | PPO | Ours ($m = 2$) | Ours ($m = 3$) |
|---|---|---|---|
| GPT 4.1 Nano Win Rate | 57.0% | 62.9% | 70.4% |

Table 7: Evaluation with feedback from GPT 4.1 nano instead of the gold reward model

### E.6 Output diversity

To quantitatively investigate how our method affects the output diversity of the trained model, we follow the methodology of Kuhn et al. (2023) to cluster outputs into semantically equivalent clusters with a deberta-large model (He et al., 2021). We use a subset of 256 prompts from the Alpaca-Farm validation split, generate 20 outputs for each with the trained Qwen 2.5 1.5B model and report the number of found clusters in Table 8. The results show a slight decrease in output diversity.

### E.7 Runtime

Our method performs well compared to standard PPO-RLHF, but the additional RM retraining also requires additional computation. We thus illustrate the additional time cost in Table 9 when training the Pythia-1B-dedup base model using 8xA100-40GB. The runtime is dominated by the PPO training steps due to the need to autoregressively generate new completions. In comparison, retraining the RM training is relatively inexpensive, first requiring around 10 minutes to calculate the importance weights and then training for 42 minutes.

| Method | PPO | Ours ($m = 2$) | Ours ($m = 3$) |
|---|---|---|---|
| # semantic clusters | 11.90 | 11.72 | 11.31 |

Table 8: Average number of semantic output clusters among 20 samples of a Qwen 2.5 1.5B model on the Alpaca-Farm validation split.

| Method | RM1 | PPO | RM2 | PPO2 | RM3 | PPO3 | Total |
|---|---|---|---|---|---|---|---|
| PPO | 42 | 507 | - | - | - | - | 549 |
| PPO+OCRM | 42 | 169 | 52 | 169 | 52 | 169 | 653 |

Table 9: Runtime (minutes) of our method and standard PPO.

## E.8  Longer DPO and WPO training

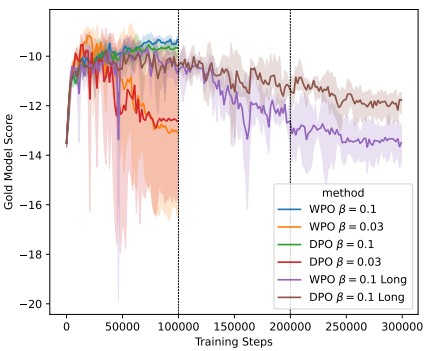

Figure 8: Performance of DPO and WPO degrades when training for longer durations.

In the main text we are showing results for training WPO and DPO for 100,000 training samples. This leads to WPO and DPO seeing fewer samples during training than the reward models used in PPO/RLP-SPG and our proposed method, making the comparison superficially unfair. We thus also attempted to train DPO and WPO on the full dataset, but we found this to lead to worse results, as shown in Figure 8. We note that this issue is consistent with findings of Xu et al. (2025, Appendix A.1). A possible explanation could be that DPO and WPO overfit to the training prompt dataset, however verifying this is outside the scope of our work.

## E.9  Sample outputs

We provide some sample input-output pairs in tables 10, 11 and 12. Note that as we are using the synthetic gold reward setup, we care more about the gold reward $R_{\text{gold}}$ than actual human preferences to evaluate the methods.

| Content | $R_{\text{gold}}$ |
|---|---|
| SUBREDDIT: r/relationships
TITLE: I [22 F] want to buy my bf [28 M] of 6 months a $500 very rare video game for his collection for his birthday/Christmas. Will a gift of this caliber scare him off/ be considered too much after being laid off from his game dev job?
POST: My boyfriend's birthday is December 24th, which is Christmas Eve. He grew up in a foster home and never got great presents for either his birthday or Christmas. He often spends his birthday and Christmas alone because his foster mother is uninterested and he doesn't have a great relationship with his birth mother.
He spent Christmas with me last year, so I went out of my way to get him both a birthday present AND a Christmas present, both of which valued above $50. However, he was so thrilled to have someone to spend his birthday and Christmas with, he showered me with 5 very nice, expensive gifts. I felt terrible. Happy, but terrible.
He is a video game artist but was recently laid off from his job. He is very passionate about his video game collection. There is one game I noticed in his collection that he only has the sequel of. I asked him why because it is very unlike him. He said it was because the original was wildly expensive and he already had a hard time affording a mint condition version of the sequel.
I have so many questions. Is this too much? Would it be creepy to get him something that he never asked for? What if giving him a game gives him bad feelings about getting laid off? What if we break up before Christmas and I am stuck with the game? Would the game have the same value to his collection if he didn't earn it himself?
Any help would be great. I am torn. | - |
| **Reference**: Foster kid boyfriend with Christmas birthday and video game collection. Is it too soon to get him a $500 game for his collection, especially after he has just been laid off from his game dev job? | -4.88 |
| **PPO**: Boyfriend of 6 months has very expensive game for his collection for his birthday/Christmas. Would it scare him off/ be considered too much/ would it be creepy to get him something he never asked for? | -8.63 |
| **DPO**: boyfriend's birthday/Christmas gift of game for his collection. Would it scare him off/ be too much? Would it be creepy to buy him a game he never asked for/ would it be worth it? | -9.13 |
| **WPO**: boyfriend's birthday/Christmas gift of $500 very rare video game for his collection. Would it scare him off from getting it or would it be too much? | -11.5 |
| **RLP-SPG**: Want to buy bf of 6 months a $500 very rare video game for his collection for his birthday/Christmas. Will a gift of this caliber scare him off/ be considered too much after being laid off from his game dev job? | -9.82 |
| **Ours**: Want to buy BF of 6 months rare video game for birthday/Christmas, worried about price/value and whether gift would scare him off/ be considered too much. Help/advice. | -5.41 |

Table 10: Example output by a Pythia-1B model trained with different RLHF methods on the summarization task.

| Content | $R_{\text{gold}}$ |
|---|---|
| What is the difference between machine learning and deep learning? | - |
| **Reference**: Machine learning is a subset of artificial intelligence that uses algorithms to learn from data and improve performance on a task without explicit human programming. Deep learning is a type of machine learning that uses artificial neural networks to learn from large amounts of data and make predictions. Deep learning is more powerful and accurate than traditional machine learning, but also requires more computing power. | -4.88 |
| **PPO**: Machine learning is a subset of artificial intelligence that uses algorithms to enable computers to learn from data and make predictions or decisions without being explicitly programmed. Deep learning is a subset of machine learning that uses neural networks to learn from data and make predictions or decisions. Deep learning is more complex than machine learning and uses more advanced algorithms to learn from data. | 1.64 |
| **DPO**: Machine learning is a type of artificial intelligence that allows a computer to learn from data without being explicitly programmed. Deep learning is a subset of machine learning that uses neural networks to learn from data and is able to learn more complex tasks. The main difference is that deep learning uses neural networks to learn more complex tasks and is able to learn from large amounts of data. | 3.44 |
| **WPO**: Machine learning is a type of artificial intelligence that allows a computer to learn from data without being explicitly programmed. Deep learning is a subset of machine learning that uses neural networks to learn from data and is able to learn more complex tasks. The main difference is that deep learning uses neural networks to learn more complex tasks than machine learning. | 3.10 |
| **RLP-SPG**: Machine learning is a subset of artificial intelligence that focuses on developing algorithms that can learn from data and make predictions or decisions based on that data. Deep learning is a subset of machine learning that uses neural networks to learn from data and make predictions or decisions. Deep learning is more advanced than machine learning and can learn more complex patterns and relationships in data. | 0.82 |
| **Ours**: Machine learning is a broader field that includes techniques such as supervised, unsupervised and reinforcement learning. Deep learning is a subset of machine learning that uses neural networks with multiple layers to learn complex patterns in data. The main difference is that deep learning uses neural networks with multiple layers to learn more complex patterns and is more computationally intensive than traditional machine learning techniques. | 6.69 |

Table 11: Example output by a Qwen 2.5 1.5B model trained with different RLHF methods on the truncated alpaca dataset.

| Content | $R_{\text{gold}}$ |
|---|---|
| SUBREDDIT: r/AskReddit
TITLE: Reddit, I think our family dog just consumed a significant amount of chocolate. What do I do?
POST: Okay so my father and I went out to dinner tonight and left the dog at home. I'd left half a bar of dark chocolate (about 1.5 oz, still in the wrapper) sitting on my desk. Mind you, the bar has been sitting on the desk for about three days now and the dog hasn't done anything about it, and typically doesn't. She knows not to climb on surfaces and has shown no interest in the candy before.
However, I returned home to find my bedroom in shambles. The contents of my desk were all knocked around and a bunch of family photos were all shredded and chewed up. I found the empty chocolate bar wrapper on the ground and still can't find the offending candy. I can only conclude that the dog has eaten it.
This is something that my dog would've done as a puppy. However, she's ten now and hasn't behaved like this in years. She's acting fine at the moment, but I'm still worried. My father doesn't seem concerned about her at all, which worries me even more. FYI, my dog is a dachshund mix and weighs about 25-30 pounds. Has anybody been in this situation before? What happened? Does anybody know what to do? | - |
| **Reference**: I think dog got into chocolate when I was out. My dad isn't concerned but I'm afraid she's going to keel over and die and I don't know what to do. | -12.7 |
| **PPO**: Dog ate chocolate bar, now my desk is destroyed and my father is not concerned. What do I do? | -9.43 |
| **DPO**: Dog ate chocolate bar that I left out for her. She's not acting like she's acting normally. What do I do? Any advice? | -9.75 |
| **WPO**: Dog ate chocolate bar, my dog has never acted like this before. What do I do? What do I do? What do I do? | -13.7 |
| **Ours**: family dog ate chocolate bar, now my room is wrecked and my father is not concerned. What do I do? Any advice appreciated. | -7.06 |

Table 12: Example output by the Pythia-2.8B model trained with different RLHF methods on the summarization task.

