# OpenReview forum: "Off-Policy Corrected Reward Modeling for Reinforcement Learning from Human Feedback"
_colmweb.org/COLM/2025/Conference — COLM 2025_

### Official Review · Reviewer_ruLM · 2025-05-07

**Rating:** 7
**Confidence:** 4
**Ethics Flag:** 1

**Summary:**

The paper proposes a new method for correcting the reward model iteratively without access to new labels or data points in RLHF. The method is based on importance weighting to scale the reword model loss. The weight is computed as the ratio between the probability returned by the policy at iteration i and the policy at iteration 1. Experimental settings with two LLM families and two standard datasets show that the proposed method improves performance while reducing the overoptimization problems of RLHF. The authors provide a substantial analysis of the methodology including ablations.

**Questions To Authors:**

Minor commnets:

* Perhaps you can improve the way you present the IW and how the ratio is calculated by making it a bit simpler (and easier to understand).

**Reasons To Accept:**

* The paper presents a simple yet effective method for tackling the overoptimization problem in RLHF.
* The method works well across LLM families and tasks. The experimental setting is good. Larger models have not been used but that's OK.
* Very good analysis and discussion of the methodology.

**Reasons To Reject:**

* N/A. Really enjoyed reading the paper.

---

> ### Author Response · Authors · 2025-05-31
>
> We would like to thank the reviewer for their helpful and kind review!
>
> We will reconsider the presentation of the importance weights and extend the algorithm in the appendix, to prevent any misunderstandings.
>
> We will also share our implementation after acceptance, to prevent any potential issues with reproducibility.

---

### Official Review · Reviewer_FVSR · 2025-05-12

**Rating:** 8
**Confidence:** 3
**Ethics Flag:** 1

**Summary:**

The paper applies a technique to progressively update the reward model being used in RLHF. The authors explain, and show via a small example, how the reward model increasingly becomes less accurate as the policy is updated. They show that using a correction to the reward model enables a better final policy to be found, as judged by a gold reward model.

**Questions To Authors:**

How far off the accuracy of the "gold reward model" were the models being used here? Did I miss this or was it not commented on? Would be helpful to know.

**Reasons To Accept:**

Clear explanation of the reward model drift issue, and benefits of using the importance weighting updates to the reward model are explained and demonstrated to have empirical benefits. Comparisons against relevant recent RLHF flavours are given, and these are done on public datasets. The showing of the results for different seeds is helpful, and the best performance of alternative methods in the comparisons is fair (using their best result, despite these dropping off in performance with continued training).

**Reasons To Reject:**

No significant reasons. Some minor comments below.
Retraining the reward model after each policy update is acknowledged as not feasible computationally, and the authors provide a pragmatic iterative approach of doing policy updates, followed by updating the reward model, and repeat `m` times. This isn't a reason to reject, just to note a limitation, which is overcome without any theoretical justification, but which empirically does prove to be helpful. It's unclear from the ablations why further iterations beyond m=3 were not done given the performance continues to improve?

---

> ### Author Response · Authors · 2025-05-31
>
> We would like to thank the reviewer for their helpful and kind review.
>
> * More iterations beyond m=3
>
> We initially used m=3 due to computational constraints.
> We now ran an additional experiment with m=4 and m=5 on Pythia 1B on the TL;DR task and indeed found a further improvement for m=4, but it seems to saturate for m=5:
>
> |             | PPO   | Ours (m=2) | Ours (m=3) | Ours (m=4) | Ours (m=5) |
> |-------------|-------|------------|------------|------------|------------|
> | Winrate     | 59.7% | 68.6%      | 71.7%      | 74.4%      | 74.9%      |
> | Gold Reward | -8.62 | -7.56      | -7.28      | -6.98      | -6.91      |
>
>
> We will add this experiment to the paper, thank you for your suggestion!
>
> * Accuracy of the Gold Reward Model
>
> When generating the reward model dataset we follow the methodology of Gao et al. [1] and always choose the completion with a higher gold RM score as preferred completion, the Gold RM thus has a perfect accuracy.
>
> For comparison we show the reward model accuracies on the RM training dataset here:
>
> | Dataset/Model | TL;DR Pythia 1B | TL;DR Pythia 2.8B | AlpacaPythia 1B | AlpacaQwen 2.5 1.5B |
> |---------------|-----------------|-------------------|-----------------|---------------------|
> | RM Accuracy   | 69.8%           | 74.4%             | 76.0%           | 77.7%               |
> | Gold Reward   | -8.62           | -7.56             | -7.28           | -6.98               |
>
> Please let us know if there is any other information about the reward models that would be helpful!
>
> References:
>
> [1] Gao et al. “Scaling Laws for Reward Model Overoptimization”, https://arxiv.org/abs/2210.10760

---

### Official Review · Reviewer_fvYh · 2025-05-17

**Rating:** 5
**Confidence:** 4
**Ethics Flag:** 1

**Summary:**

The paper proposes Off-Policy Corrected Reward Modeling, a method that addresses overoptimization in RLHF. It aims to correct the distribution shift between the SFT and the updated RL policy and apply importance weighting (IW). The authors provide a theoretical justification, derive estimation error bounds, and present experiments on summarization and chatbot tasks using simulated gold reward models.

**Reasons To Accept:**

1. It proposes a theoretically grounded and practically implementable method using off-policy correction via importance weighting.

2. Empirical results compared to PPO, DPO, WPO, and RLP-SPG show the effectiveness of proposed IW methods.

3. The paper is well structured and easy to follow.

**Reasons To Reject:**

1. It only tested on two tasks (TL;DR summarization and Alpaca-Farm chatbot), both with synthetic gold RMs rather than human annotators or real-world alignment benchmarks.

2. This is no downstream or qualitative analysis on reward changes. It is unclear how changes in reward scores reflect actual quality, diversity, or safety of generated outputs.

3. Experiments are only conducted on small models, about 1B. Larger sizes of models are required to demonstrate its generalization and improvement.

---

> ### Author Response · Authors · 2025-05-31
>
> We would like to thank the reviewer for their thoughtful and helpful review.
>
>  * Only two tasks and synthetic Gold RM setting
>
> We acknowledge that this is a limitation. We decided to focus on fewer datasets and models in order to provide more complete ablations instead, which we hope to provide a better understanding of our method.
> We unfortunately do not have the resources to perform training runs with real human feedback instead of a synthetic setting.
>
>  * Qualitative analysis of effect on outputs
>
> We performed an additional analysis to investigate the diversity of the model outputs, based on the methods suggested in Kuhn et al. [1]:
> On a subset of 256 prompts from the validation set, for each prompt, we sample 20 completions and use deberta-large to cluster them into semantic clusters. We report the average number of clusters. The results show that our method maintains a similar output semantic diversity compared to the PPO baseline:
>
> | Qwen 1.5B Alpaca | PPO   | Ours (m=2) | Ours (m=3) |
> |---------------------|-------|------------|------------|
> | # semantic clusters | 11.90 | 11.72      | 11.31      |
>
> We will further extend this experiment along with more example outputs and add it to our submission, thank you for the suggestion!
>
> We also provide statistics of the gold reward distribution on the validation set, indicating that the improvement is not limited to specific prompts:
>
> | Qwen 1.5B Alpaca |  STD | 25th percentile | Median | 75th percentile |
> |------------------|-----------------|-----------------|--------|-----------------|
> | PPO              | 8.23            | -1.45           | 4.39   | 9.75            |
> | Ours (m=2)       | 8.04            | -0.48           | 5.34   | 10.06           |
> | Ours (m=3        | 7.94            | -0.25           | 5.80   | 10.31           |
>
>
>  * Model size limited to 1B
>
> We would like to note that we also ran experiments on Pythia 2.8B, shown in Table 3 in the submission and reproduced here, which shows that our method still is beneficial with increased model sizes:
>
> | Pythia 2.8B | PPO   | Ours (m=2) | Ours (m=3) |
> |-------------|-------|------------|------------|
> | Winrate     | 47.3% | 63.4%      | 75.4%      |
>
> We understand that this is still not comparable to large-scale model training runs, unfortunately we do not have the computational resources to scale the experiments to large LLMs.
>
>
> Thank you for the suggestions, we hope that we could address your concerns.
> We would be grateful for additional suggestions on specific qualitative analyses for the output.
>
> References:
>
> [1] Kuhn et al. “Semantic Uncertainty: Linguistic Invariances for Uncertainty Estimation in Natural Language Generation”, ICLR 2023

---

> ### Author Response · Authors · 2025-06-03
> **Additional GPT-labeled experiment**
>
> To address the reviewer's concerns about the synthetic gold model setup we ran an additional experiment with an API model as a replacement for the gold model.
> We train Pythia-1B on the TL;DR summarization task with pairwise comparisons provided by GPT 4.1 Nano, both to create the preference labels for the RM training dataset and for the final evaluation.
>
> The results are consistent with the gold reward model experiments shown in the paper:
> | Pythia 1B            | PPO   | Ours (m=2) | Ours (m=3) |
> |----------------------|-------|------------|------------|
> | GPT 4.1-Nano Winrate | 57.0% | 62.9%      | 70.4%      |
>
> We will add this experiment to the paper and hope it can address the reviewer's concerns about our experiments. Thank you for the suggestion!

---

> ### Author Response · Authors · 2025-06-07
> **Additional 6.9B experiment**
>
> To address the reviewer's concerns about the model size, we now added an additional experiment with GRPO to train a Pythia 6.9B model on the summarization task:
>
> | Pythia 6.9B | GRPO  | Ours (m=2) | Ours (m=3) |
> |-------------|-------|------------|------------|
> | Winrate     | 64.8% | 78.6%      | 79.6%      |
>
> The results show that our method also works for larger models.
>
> We would like to thank the reviewer for their suggestion and hope that that we could address their concerns about the experimental evaluation.

---

### Official Review · Reviewer_4P8x · 2025-05-26

**Rating:** 7
**Confidence:** 4
**Ethics Flag:** 1

**Summary:**

This paper proposed a simple yet effective method to train reward models in RLHF to mitigate the distribution shift issue. The method applies an importance weighting factor in the loss function of reward model training, where the importance weighting factor is calculated by the current policy and the SFT policy. The effectiveness of the method is verified empirically. Sufficient ablation studies are provided, offering informative insights into the problem and method.

**Questions To Authors:**

- Line 3, "online RLHF": The term "online RLHF" is used multiple times in the paper with a reference to Stiennon et al., 2020. However, to the best of my knowledge, this term is not well-established. In fact, it seems somewhat misleading, as neither the RLHF method presented in this paper nor the method proposed by Stiennon et al., 2020, are truly "online." Both approaches involve collecting pairwise responses using a policy and annotating preference labels with human (gold) labelers. In lines 72-73, it is stated, "While they focus on offline RLHF, we focus on online RLHF." I recommend providing clear definitions or appropriate references if the terms "online RLHF" and "offline RLHF" are to be used, or directly using the term "RLHF" instead.
- Line 170 and Equation 5: Could you explain the meaning of $P_{\pi^i}(s, a_w, a_l)$? Why does the equation $P_{\pi^i}(s, a_w, a_l) = P(s) \pi^i(a_w | s) \pi^i(a_w | s) P(a_w > a_l | s)$ hold?

**Reasons To Accept:**

- The issue and method are presented clearly with precise formulations and clear language.
- The proposed method is both simple and effective.
- The empirical experiments are thorough and comprehensive.

**Reasons To Reject:**

- The clarity of the presentation requires improvement; refer to the questions for further details.
- The results reported in Table 1 appear incomplete. Results for both tasks and both base models should be included for a more comprehensive comparison. Reporting results for each random seed is unnecessary. Providing the mean and standard deviation would suffice.

---

> ### Author Response · Authors · 2025-05-31
>
> We would like to thank the reviewer for their helpful advice and questions.
>
> We will revise Table 1 and move the results for individual random seeds to the Appendix.
>
> * “Online RLHF” terminology
>
> While we note that the terminology has been used in recent papers [1,2], we agree that it can be misleading as no new preference data is obtained online in our problem setting, instead only new completions are sampled online from the policy. We will revise the relevant sections to replace this terminology.
>
> * Meaning of $P_{\pi^i}(s,a_\mathrm{w},a_\mathrm{l})$
>
> We consider a data generation process in which we first sample the prompt $s \sim P(s)$, then independently sample two completions from the model $\pi^{i}$,  $a_1\sim \pi^i(a|s)$, $a_2 \sim \pi^i(a|s) $ and finally sample a human preference label $y \sim P(y|s,a_1,a_2)$, which is 1 if $a_1$ is preferred over $a_2$ and 0 otherwise.
> Together, this yields the joint probability
>
> $P_{\pi^i}(s,a_1,a_2,y) = P(s)\pi^i(a_1|s)\pi^i(a_2|s)P(y|s,a_1,a_2) .$
>
> In the current version of our submission the preference label $y$ is absorbed into the ordering of terms $a_\mathrm{w}$ and $a_\mathrm{l}$, respectively for the winning and losing reply.
> We hope this explanation clarifies the term and we will improve the presentation in the submission.
>
> We believe these changes will improve the clarity of our submission, thank you!
>
> References:
>
> [1] Dang et al., “RLHF Can Speak Many Languages: Unlocking Multilingual Preference Optimization for LLMs”, EMNLP 2024
>
> [2] Ethayarajh et al., “KTO: Model Alignment as Prospect Theoretic Optimization”, ICML 2024

---

> > ### Comment · Reviewer_4P8x · 2025-06-01
> > **Additional experimental results**
> >
> > Thank you for your response. Most of my previous questions were clarified. However, my concerns regarding the experimental results still remain, as outlined in my previous comments. It seems that these concerns are aligned with feedback from other reviewers.

---

> > > ### Author Response · Authors · 2025-06-06
> > > **Requested experimental results**
> > >
> > > We now also evaluated our method on Qwen 2.5 1.5B on the TL;DR task, it also shows a notable improvement over standard PPO:
> > >
> > > | Qwen 2.5 1.5B TL;DR | PPO   | Ours (m=2) | Ours (m=3) |
> > > |---------------------|-------|------------|------------|
> > > | Winrate             | 80.5% | 87.7%      | 89.6%      |
> > > | Gold Reward         | -6.06 | -5.04      | -4.68      |
> > >
> > > We believe this makes the results more complete, now evaluating both base models on both tasks as you suggested.
> > >
> > > We will add it to the paper, thank you!

---

> > > > ### Comment · Reviewer_4P8x · 2025-06-06
> > > > **Thank for the additional results**
> > > >
> > > > Thank you to the authors for providing additional results. My concerns have been satisfactorily addressed.

---

### Author Response · Authors · 2025-06-07
**Summary of Additional Experiments**

We would like to summarize the main improvements we made and new experiments we performed during the rebuttal here.

 * Evaluation with feedback from API LLM:

Reviewer fvYh raised concerns about the gold RM setup used in our experiments. We now added an experiment which uses GPT 4.1 Nano instead of the gold RM and is consistent with the results of the gold model setup presented in the paper:

| Pythia 1B            | PPO   | Ours (m=2) | Ours (m=3) |
|----------------------|-------|------------|------------|
| GPT 4.1-Nano Winrate | 57.0% | 62.9%      | 70.4%      |

 * Added 6.9B model:

Reviewer fvYh raised concerns about the experiments being focused on 1B models. We note that the submission includes a 2.8B experiment, and now added a new 6.9B model:

| Pythia 2.8B | PPO   | Ours (m=2) | Ours (m=3) |
|-------------|-------|------------|------------|
| Winrate     | 47.3% | 63.4%      | 75.4%      |

| Pythia 6.9B | GRPO   | Ours (m=2) | Ours (m=3) |
|-------------|-------|------------|------------|
| Winrate     | 64.8% | 78.6%      |    79.6%     |



 * Longer Training

Reviewer FSVR suggested extending the experiments with more iterations, we thus added experiments for m=4 and m=5, which shows an additional improvement before saturation

| Pythia 1B | PPO   | Ours (m=2) | Ours (m=3) | Ours (m=4) | Ours (m=5) |
|-----------|-------|------------|------------|------------|------------|
| Winrate   | 59.7% | 68.6%      | 71.7%      | 74.4%      | 74.9%      |

 * Output Evaluation

Following reviewer fvYh’s suggestion we added an evaluation of the diversity of outputs of methods, showing a similar diversity of outputs of our method compared to baseline PPO

| Qwen 1.5B Alpaca    | PPO   | Ours (m=2) | Ours (m=3) |
|---------------------|-------|------------|------------|
| # semantic clusters | 11.90 | 11.72      | 11.31      |


 * Qwen Evaluation on TL;DR

Reviewer 4P8x suggested evaluating Qwen on both the summarization and chatbot dataset, as initially we only evaluated it on the chatbot task. The results also show an improvement when using our method.

| Qwen 2.5 1.5B TL;DR | PPO   | Ours (m=2) | Ours (m=3) |
|---------------------|-------|------------|------------|
| Winrate             | 80.5% | 87.7%      | 89.6%      |

We believe these additional experiments strengthen the evaluation of our method and hope that they could address the reviewer’s concerns.

We would like to again thank all reviewers for their advice!

---

### Decision · Program_Chairs · 2025-07-08

**Decision:**

Accept

**Comment:**

This paper addresses overoptimization in RLHF, where the policy exploits inaccuracies in the reward model arising from distribution shift. The authors frame this issue as an inconsistent estimation of the reward model parameters, which is insightful. Their proposed solution, to iteratively retrain the reward model using off-policy correction with importance weighting, is well-motivated. The empirical evidence on summarization and chatbot datasets demonstrates a clear and significant improvement over existing methods. Reviewer fvYh raised concerns around the synthetic settings of the experiments and generalization of the findings to larger models. The authors provided additional results showing benefits in more realistic and larger model settings. The paper is well-written, the approach is sound, and the results are compelling, making a valuable contribution to the field.